

# Modeling the Potential Impacts of Climate Change on the Hydrology of Selected Forested Wetlands in the Southeastern United States

Jie Zhu[1,2,3], Ge Sun[4*], Wenhong Li[3*], Yu Zhang[3], Guofang Miao[5,6], Asko Noormets[6], Steve G. McNulty[4], John S. King[6], Mukesh Kumar[3], Xuan Wang[1,2]

[1]State Key Laboratory of Water Environment Simulation, School of Environment, Beijing Normal University, Beijing 100875, China
[2]Key Laboratory for Water and Sediment Sciences of the Ministry of Education, School of Environment, Beijing Normal University, Beijing 100875, China
[3]Nicholas School of the Environment, Duke University, Durham, North Carolina 27708, USA
[4]Eastern Forest Environmental Threat Assessment Center, USDA Forest Service, Raleigh, North Carolina 27606, USA
[5]Department of Natural Resources and Environmental Science, University of Illinois at Urbana-Champaign, Illinoi 61801, USA
[6]Department of Forestry and Environmental Resources, North Carolina State University, Raleigh, North Carolina 27695, USA

*Correspondence to*: Ge Sun (gesun@fs.fed.us) or Wenhong Li (Wenhong.li@duke.edu)

**Abstract.** Riverine floodplains and coastal margins of the southeastern United States host extensive forested wetlands, providing myriad ecosystem services including carbon sequestration, water quality improvement, groundwater recharge, and wildlife habitat. However, these ecosystems, which are closely dependent on wetland hydrology, are at risk due to human-made climate change. This study develops site-specific empirical hydrologic models for five forested wetlands with different characteristics by synthesizing long-term observed meteorological and hydrological data. These wetlands represent typical Cypress Ponds/Swamps, Carolina Bays, Pine Flatwoods, and Wet Pine, and natural Bottomland Hardwoods ecosystems. The validated empirical models are then applied at each wetland to predict future water table changes using climate projections from 20 General Circulation Models (GCMs) participating in the Coupled Model Inter-comparison Project 5 (CMIP5) under both Regional Concentration Pathways (RCP) 4.5 and RCP 8.5 greenhouse gas emission scenarios. We show that projected combined changes in precipitation and potential evapotranspiration would significantly alter wetland groundwater dynamics in the 21st century. Compared to the historical period, all five studied wetlands are predicted to become drier by the end of this century. The water table depth increases vary from 4 cm to 22 cm due to global warming. The large decrease in water availability (i.e., precipitation minus potential evapotranspiration) will cause a drop in the water table in all the five studied wetlands by the late 21st century. Among the five examined wetlands, the depression wetland in hot and humid Florida appears to be most sensitive to climate change. This modeling study provides quantitative information on the potential magnitude of wetland hydrological response to future climate change for typical forested wetlands in the southern U.S. Study results suggest that the ecosystem functions of southern forested wetlands will be substantially impacted by future climate change due to hydrological changes that are the key control to wetland biogeochemical cycles, vegetation distribution, fire regimes, and wildlife habitat. We conclude that climate change assessment on wetland





forest ecosystems and adaptation management planning in the southeastern U.S. must first evaluate the impacts of climate change on wetland hydrology.

## 1 Introduction


The importance of the hydrology of forested wetlands in regulating ecosystem functions has long been recognized (*Amatya et al., 2006; Sun et al., 2002; Sun et al., 2000)*. Wetlands provide critical ecosystem services such as groundwater recharge, water quality improvement, flood control, carbon sequestration, wildlife habitat, and recreation (*Hammack and Brown,* 2016*; Greenberg et al.,* 2015*; Richardson,* 1994). Wetland hydrology plays an important role

in biogeochemical cycles such as the emission of greenhouse gases of $CH_4$, $CO_2$, $NO_x$ (*Dai et al.,* 2013; *Zhang et al.,* 2012) and therefore has an influence on regional and global climate (*Paschalis et al.*, 2017). A small change in wetland water level, even by less than 10 cm, may have profound impacts on wetland structure and other ecosystem functions (*Webb and Leake*, 2006).

Wetland hydrology is strongly influenced by the variation and change in climate (*Brooks*, 2009; *Fossey and*

*Rousseau*, 2016; *Liu and Kumar*, 2016). Regional wetland area losses are predicted in the United States and globally (*House et al.* 2016; *Nicholls*, 2004) under future climate change. Wetland hydrology is extremely dynamic in space and time and is sensitive to climate change (*Bullock and Acreman*, 2003; *Short et al.*, 2016), especially in the southeastern United States (SE US) (*Li et al., 2013*; *Li and Li*, 2015; Dai et al., 2013; Dai et al., 2010; Lu et al. 2010). Evidence of climate change in the region is plenty and climate models project that the temperature will increase by

2 °C–10 °C by 2100 in this region (*Diffenbaugh and Field*, 2013). The severity and patterns of storms are changing as well, with more heavy downpours in many parts of the SE US, and more powerful Atlantic hurricanes landfall (Wang et al., 2010).

However, process-based study on the impacts of climate change on different wetlands in the SE US are limited. Various hydrological models, ranging from regression models to complex distributed models, have been used to study

hydrological response to climate change. For example, the physically based distributed model MIKE SHE has been applied to forested wetlands in the SE US (Dai et al., 2010; Lu et al., 2009; House et al., 2016). The hydrological regime of wetland forests on the coastal plains of South Carolina was found to be highly sensitive to annual precipitation and temperature changes (Dai et al., 2010). The water table of pine flatwoods in Florida was predicted to be 20-40 cm lower than the baseline scenario when precipitation decreased by 10 % or temperature increased by

2 °C (Lu et al., 2009).

Hydrological models have been widely used to study interactions among climate, water, and vegetation (*Sun et al.,* 1998). Although physically based hydrological models provide refined understanding of hydrologic processes (Yu et al., 2015; Chen et al., 2015) and detailed estimates of hydrologic states and fluxes (Qu and Duffy, 2007; Shen and Phanikumar, 2010), these models are generally data (Bhatt et al., 2014) and computation intensive (Vivoni et al.,

2011), their potential uses are often undercut by equifinality of parameters (*Beven*, 1993; *Kumar et al.*, 2013; *Pokhrel et al.*, 2008). Implementing distributed hydrologic models at multiple wetlands that cover a range of climatic, topographic, and management conditions is challenging because of the computational expense, lack of fine scale input

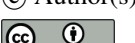



data, and poor adaptability of model parameters (*Grayson et al.*, 1992). In contrast to physically based distributed hydrological models, statistical models developed based long-term empirical data have been demonstrated to have
advantages of high efficiency over physically based models. Performance such type of models in climate change studies appears to be satisfactory when incorporating downscaled Global Climate Model (GCM) outputs (*Sachindra et al.*, 2013; *Li et al., 2016*). For example, *Li et al.* (2016) successfully used log-linear models for 21 rainfall stations and 7 hydrometric stations to predict hydrological drought. Greenberg et al. (2015) developed an empirical model and applied the model to study the impacts of climate change on wildlife habitat.

In this study, we modelled the potential hydrological responses to climate change for five forested wetlands covering a range of climatic, topography and management conditions in the southeastern U.S. Future climate data from all the 20 GCMs participating in the Coupled Model Inter-comparison Project 5 (CMIP5) under both Regional Concentration Pathways (RCP) 8.5 and RCP 4.5 scenarios were used. We hypothesized that the wetlands would become drier due to climate warming and increased in evapotranspiration. We also hypothesized that hydrological
responses would vary due to differences in background climate and wetland physical configurations.

    The objectives were to: 1) construct and validate empirical models of wetland groundwater dynamics using long-term observational data in five forested wetlands, 2) forecast water table changes in the five wetlands under climate change under 40 scenarios (i.e., 20 GCMs and two $CO_2$ emission pathways) s, and 3) investigate the key mechanisms driving the impacts of climate change in each wetland.

**2 Methods**

**2.1 Study area**

    We selected five long-term research sites in the SE US representing five types of wetlands with a different combination of climate, topography and anthropogenic management disturbances. These research sites include (1) Alligator River Wildlife Refuge bottom hardwood wetland (designated as AR) on the coast of North Carolina, (2) a
drained wetland converted from pocosin wetland to pine plantation forest (LP) on the lower coastal plain of North Carolina, (3) cypress pond wetland (wetland FL–WET) in north central Florida, (4) upland slash pine forest (wetland FL–UP) in northern central Florida, and (5) Carolina Bays (SC) on the coastal plain of South Carolina (Fig. 1). The wetland characteristics (e.g., climate, soil, vegetation, wetland type classification) have contrasting features (Table 1). These wetlands were selected with the following considerations. The AR (*Miao*, 2013) and LP (Noormets et al., 2010;
Sun et al., 2010; *Tian et al.*, 2015) are located in the lower coastal plain area of North Carolina with 62 miles apart from one another, representing the lower coastal plain forested wetlands with similar climate and topography but different management conditions. The AR is a natural coastal bottomland hardwoods wetland with no astronomic tides (*Miao et al.*, 2013), while wetland LP is artificially managed by the forestry industry for timber production (*Manoli et al.*, 2016; Noormets et al., 2010; *Sun et al.*, 2010). LP is located within the outer coastal plain mixed forest province of North Carolina. The area has been artificially drained with a network of field ditches (90–100 cm deep; spacing
80–100 m) and canals dividing the watershed into a mosaic of regularly shaped fields and blocks of fields (*Sun et al.*,


2010). FL–WET and FL–UP (*Lu*, 2006; *Lu et al.*, 2009) represent two types of ecosystems found the same pine flatwoods landscape with the same climate but slight different elevation and management. FL–UP is dominated by slash pine (*Pinus elliotii*) plantation forests on a relative higher elevation while FL–WET is dominated by naturally

generated cypress (*Taxodium distichum*) on depression areas in pine flatwoods. The FL research site is located 33 km northeast of Gainesville in the Alachua County of north central Florida. The SC wetland was located in Bamberg County, South Carolina representing a typical depression wetland in the region (Dai et al., 2010; Dai et al., 2013; *Pyzoha et al.*, 2008; *Sun et al.,* 2006) The SC wetland was covered by naturally regenerated deciduous trees (i.e., water oak, willow oak) and was surrounded by deep, well-drained sand dominated by hardwood plantations and agricultural

crops (Dai et al., 2010; Dai et al., 2013; *Pyzoha et al.*, 2008; *Sun et al.*, 2006).

## 2.2 Databases

### 2.2.1 Observed water table and meteorological data

The data details and the collection methods used in this study are summarized in Table 2. The meteorological variables include the precipitation, air temperature, wind speed, net radiation, and other canonical meteorological

factors. The daylight duration data were from The United States Naval Observatory (USNO) (http://aa.usno.navy.mil/data/docs/Dur_OneYear.php). The dataset consists of 48,826 30-min time series observations for each variable (i.e., water table and meteorological variable) for AR, 2,922 daily time series observations for LP, and 89,121 daily time series future climate data for each variable from each GCM of all five sites. The 30-min air temperature was averaged at the daily scale for estimating the potential daily evapotranspiration using Hamon's

equation (*Federer and Lash*, 1978a; *Hamon*, 1963):

$$PET = 29.8 \times D \times \frac{e_a^*}{T_a + 273.2} \tag{1}$$

where PET is potential daily evapotranspiration (mm/day), D is day length (hr), and $e_a^*$ is the saturation vapor pressure (kPa) at the mean daily air temperature (AT, ℃) calculated by the equation modified from Dingman (2015):

$$e_a^* = 0.611 \times \exp\left(\frac{17.3 \times AT}{AT + 237.3}\right) \tag{2}$$

A correction coefficient (Sun et al., 2002) was multiplied to adjust PET calculated by Hamon's equation to derive realistic PET values for forests. The correction coefficients were reported to range from 1.0 to 1.2 (North Carolina, Federer and Lash, 1978b), and was 1.3 for the Florida site (Sun et al., 1998), thus to reduce uncertainty of the coefficient, the correction value of 1.2 was multiplied for all studying wetlands in SE US in this study.

### 2.2.2 Future climate change data

The daily climate data were derived from the 20 GCMs for two future RCP scenarios (RCP 4.5 and RCP 8.5; 2006–2099). Future climate data represent intermediate and high greenhouse gas (GHG) emission scenarios with respect to a historical climate forcing baseline (1950–2005) (*Duan et al.*, 2016).

The data of the 20 GCMs were obtained from the Multivariate Adaptive Constructed Analog (MACA) dataset, which was statistically downscaled from the native resolution of the GCMs of the CMIP5 to either 4- or 6- km





(*Abatzoglou and Brown*, 2012) (http://maca.northwestknowledge.net/index.php). We analyzed future climate conditions key to wetland hydrology, including the daily maximum temperature near the surface (2 m), daily minimum temperature near the surface (2 m), and projected average daily surface precipitation from January 1, 2050, to December 31, 2099. Daily maximum and minimum air temperatures were averaged to derive daily air temperature (*Klein et al.*, 2002). To analyze the historical and future hydroclimatic change, we selected three 20-yr time periods

according to IPCC Assessment Report 5 (2014): end of the 20th century (1980–1999), future mid-21st century (2040–2059), and end of 21st century (2080–2099). The five climate scenarios used in the paper are:

    i.    Scenario B for baseline period 1980–1999 (baseline scenario);

    ii.    Scenario F1 using RCP 4.5 for the period 2040–2059;

    iii.    Scenario F2 using RCP 4.5 for the period 2080–2099;

iv.    Scenario F3 using RCP 8.5 for the period 2040–2059;

    v.    Scenario F4 using RCP 8.5 for the period 2080–2099.

## 2.3 Model Development

The fluctuation of water table reflects the water balance between inputs (i.e., precipitation (P), groundwater and surface inflows) and outputs (i.e., outflow and evapotranspiration (ET)). Therefore, we hypothesized that these fluxes

(i.e., P, ET) and associated meteorological variables would play significant roles on water table fluctuations. For example, precipitation patterns influence the discharge of rivers and streams, affecting the frequency and duration of inundation along these waterways and adjacent wetlands (*Larsen et al.*, 2016). Water tables rise, and area of wetland expand with cooler temperatures, lower evaporation rates, and increased rainfall (*Li et al.*, 2007). The lagged 15-day mean water table (i.e., $Y_{t-1}$, $Y_{t-2}$) was also considered as potential independent variables (Greenberg et al., 2015).

A general linear model (*Greenberg et al.*, 2015; *Lydia et al.*, 2016; *Tran et al.*, 2016) was then established with all the above variables:

$$Y_{it} = \alpha_{i0} + \beta_{i1} X_{1t} + \beta_{i2} X_{2t} + \cdots + \beta_{in} X_{nt} + \gamma_{i1} Y_{it-1} + \gamma_{i2} Y_{it-2} + u_t \qquad (3)$$

where $Y$ is the water table; $X_1, \dots X_n$ are the designated matrix of individual climate variables such as total precipitation, mean air temperature, PET, total P-PET, etc.; $i$ is the number of the wetlands, $i = 1, 2, 3, 4, 5$, which

denote wetland AR, LP, SC, FL–UP, and FL–WET, respectively; t is the time period; $\alpha, \beta$ and $\gamma$ are regression parameters generated by the model; $\alpha_{i0}$ is the intercept; $\beta_{in}$ is the coefficient of the variable $X_n$ of the $i$th wetland; and $\gamma_i$ is the coefficient of $Y$ (water table of the $i$th wetland). $Y_{t-1}$ and $Y_{it-2}$ are the lagged 15-day water table of the contemporaneous water table $Y_t$.

Although water table dynamics are also affected by site-specific factors such as ditching/drainage, subsurface flow

due to topographic differences and local landscape hydrology, they were not considered explicitly in our model. For example, regarding the AR wetland, besides the climatic forcing in the model, the future water table changes will also be impacted by the changing local hydrology due to sea level rise in this area (Miao, 2013). Our main aim is to evaluate the potential impacts of climate change on wetland hydrology and the aforementioned site specific factors are assumed static We assume that the effects of these local site characteristic factors are nonetheless taken into account indirectly

by the coefficients (i.e., intercepts) of the model models.





Actual water loss from wetlands (ET) is controlled by both PET and precipitation and is a major output of water loss from a forested wetland (*Sun et al.*, 2002). Here we used an air temperature-based Hamon equation to estimate PET (*Hamon*, 1963). Also, PET, instead of air temperature, was introduced into the model since PET was affected not only by air temperature, but also daytime length, which can better reflect evaporative demand in different locations than air temperature alone.

According to the U.S. wetland regulatory standards, an area would be qualified as wetland when it is wet enough to be saturated within 1 ft (i.e., ~30 cm) of the ground surface for two weeks or more during the growing season in most years (*Tiner*, 2016). In addition, it is suggested that the water of wetlands should be held in impoundments for at least two weeks to provide weed control and also prolong wildlife use of habitat (*Nelms*, 2007). Thus, we set 15 days as the model time step and all-time series data were transformed to a15-day interval.

We then implemented correlation analysis and stepwise regression procedures to develop a parsimonious model for predicting wetland water table dynamics for each wetland. All independent variables were individually standardized first and introduced to the stepwise regression procedures to select the independent variables highly correlated to the contemporaneous water table. The correlation analysis between any two of the selected independent variables was executed to distinguish paired collinear variables. To reduce the multicollinearity, each of the paired collinear variables was removed by turns, and the other selected independent variables were then accordingly reintroduced to the stepwise regression procedures to seek a balance between the best statistical performance of the model and minimal multicollinearity of the independent variables (*Sachindra et al.*, 2013). The correlation analysis and the stepwise regression model procedures were combined in this study to obtain an optimized model with least variables and best statistical performance.

The final model was chosen based on the coefficient of determination ($R^2$) and probability (P) value at a confidence level of 95 %. Durbin's $h$ statistic (*Bhargava et al.*, 1982) was used to test for autocorrelation of the water table at a given time lag (i.e., t-1, t-2). Data were separated to two groups that cover different periods for model development and validation purposes (Table 2). For example, at wetland FL–WET, the time series including wet and dry years (1993–1994) was used to develop the model, and the remaining data (1992, 1995, and 1996) were used for model validation (Fig. 3).

For a better understanding both the variabilities of long-term averages and short-term extreme water table dynamics, the modeled future water table was analyzed at annual and 15-day scales, respectively. The 15-day lowest water table results were further analyzed for two cases: 1) the percentage of time when water table is lower than 0 cm, representing the likelihood of a wetland without surface water ponding, and 2) the percentage of time when water table depth is between 0 and -30 cm, representing the likelihood of saturated soil. This 30-cm definition was based on previous studies that suggested wetland soils have a 30-cm saturated fringe and the average root depth is about 30-cm (Tiner, 2016). The 30-cm depth was also observed as the boundary for $CH_4$ emission (*Moore and Knowles*, 1989), ammonification, denitrification (water table depth <30 cm) and nitrification (water table depth >30 cm) (*Hefting et al.*, 2004).



## 3 Model results

### 3.1 Selected models and model performance

The stepwise regression results suggest that the following linear model form best fits the water dynamics at all five wetlands:

$$Y_{it} = \alpha_{i0} + \beta_{i1} X_{1t} + \gamma_{i1} Y_{it-1} + u_t \tag{4}$$

where $X_{1t}$ is the P-PET in mm per15 days, Y is the water table depth of wetland i (i=1, 2, 3, 4, 5) in cm at time t, and t, t-1 is the current and previous time step, respectively. The statistics and parameter values for the five wetlands vary (Table 3). The predicted WT matched observed water tables consistently for all five wetlands (Fig. 2). Amongst the five wetlands, $\beta_{i1}$ and $\gamma_{i1}$ were different but generally close, ranging from 0.11 to 0.40 and from 0.77 to 0.87, respectively (Table 3), suggesting there are some site-specific differences, but t the influence of P-PET and antecedent water table at t-1 time step on the present water table at t time step is similar across the study sites. However, the intercepts $\alpha_{i0}$ vary significantly, with a maximum of 23.2 (FL–UP) and a minimum of -1.2 (AR), indicating that there may be other site specific factors that could vary across different wetlands but are not explicitly included in the model as independent variables.

The statistical models were then validated using independent subsets of water table data during the validation period (Table 2). $R^2$ values of regression between the observations and predictions were 0.77 for AR, 0.97 for LP, 0.67 for FL–UP, 0.55 for FL–WET, and 0.91 for SC, respectively (Fig. 3). The results show that the models performed reasonably well for all five wetlands during the validation years, and could be good candidates used to determine the future changes in water table due to climate change.

### 3.2 Patterns of future climate and PET

The increase of the future mean annual air temperature in RCP 8.5 is expected to be 3.9 °C, 4.3 °C, 4.0 °C, and 4.4 °C for AR, LP, FL, and SC, respectively (Table 4, Fig. S1), with respect to the historical baseline period (i.e., 1980–1999). The average increase from the baseline to RCP 8.5 in the five wetlands would be approximately 4 °C, consistent with the U.S. climate assessment report (*Pachauri et al.*, 2014). Similarly, the future annual total PET, would increase by 23 % (221 mm), 25 % (238 mm), 23 % (267 mm), and 25 % (266 mm) for AR, LP, FL, and SC, respectively, in the RCP 8.5 scenario compared with that of the historical baseline period (Table 4). The increase in PET is expected to be smaller in the RCP 4.5 scenario (Tables S1–S5, Fig. S2). For example, PET of wetland AR would increase by 13 % (130 mm) in the RCP 4.5 scenario (1107 mm), while the increase is 23 % (221 mm) in the RCP 8.5 scenario (1198 mm, Table S1).

The baseline mean annual precipitation was 1266 mm, 1275 mm, 1318 mm, 1192 mm (Tables S1-S5, Fig. S3) for AR, LP, FL, and SC, respectively. The annual total precipitation under RCP 8.5 scenario would increase the most in the wetlands LP (63 mm) and SC (60 mm) (Table 4), which is nearly two times the increase in wetland AR (37 mm). In contrast, the annual precipitation is projected to decrease at FL by 21 mm (Table 4). It is noteworthy that, unlike air temperature and PET, the magnitudes of the precipitation changes in the future RCP 8.5 scenario were smaller than



245 that of the RCP 4.5 scenario (Tables S4–S5). Specifically, the precipitation would increase by 56 mm, 68 mm, and 70 mm (Tables S1-S3) under the RCP 4.5 scenario for wetland AR, LP and SC, respectively.

Future estimated PET will increase in a larger magnitude than precipitation, causing a decrease in P-PET for all five wetlands. Specifically, the future annual mean P-PET in the RCP 8.5 scenario would decrease by 64 % (decrease by 184 mm from the 290 mm of baseline), 56 % (decrease by 175 mm from 313 mm), 175 % (decrease by 289 mm from 165 mm), and 146 % (decrease by 207 mm from 142 mm) at AR, LP, FL, and SC, respectively (Fig. 4, supplementary Tables S1–S5). The decrease in P-PET is smaller in RCP 4.5 scenario. For example, the annual P-PET at AR would decrease by approximately 75 mm (26 % of baseline) in RCP 4.5 and 184 mm (64 % of baseline) in RCP 8.5 (Table S1).

### 3.3 Future water table dynamics

### 3.3.1 Predicted annual water table

This modelling analysis suggests that future climate change may considerably affect wetland hydrology. The annual average water table exhibits a decreasing trend in all the five wetlands predicted by the 20 GCMs under both RCP8.5 and RCP4.5 scenarios (Fig. 5). In AR, the annual mean water table will decrease by 4 cm from a long term mean of 0 cm depth, Table S1) from the historical baseline period to the future RCP 8.5 scenario. It would decrease by 19 cm in LP (originally -100 cm, Table S2), by 7 cm in SC (originally -16 cm, Table S3), by 17 cm (originally -73 cm, Table S4) in FL–UP and by 22 cm (originally 2 cm, Table S5) in FL–WET.

### 3.3.2 The future 15-day water table changes

At the 15-day scale, similar to the annual change, future water table would generally decline at all sites under both RCP 4.5 and RCP 8.5, especially for RCP 8.5 scenarios (Fig. 6). For AR, the decrease of the 15-day lowest water table would be 7 cm, from -10 cm of the historical baseline period to the future -17 cm under the RCP 8.5 scenario (Fig. 6). The decrease for LP, SC, FL–UP, FL–WET would be 28 cm (from -135 cm), 14 cm (from -28 cm), 23 cm (from -101 cm), and 27 cm (from -19 cm), respectively (Fig. 6).

Additionally, all the 15-day water table records are negative (i.e., water table < 0 cm) at LP, FL–UP, and SC, meaning there is no surface water ponding over the study period from the baseline to the future RCP 8.5 scenario (Table 4, Fig. 6). In contrast, wetlands AR and FL–WET show a low probability (40 % for FL–WET, 49 % for AR) of no surface water ponding in the baseline, but increase significantly to 62 % and 93%, respectively.

While LP, FL–UP and SC are predicted to have no surface water ponding over the study period,, site LP and FL–UP would remain 0 % saturated from baseline, with a water table level lower than -30 cm based on the RCP scenario, and the saturation probability of wetland SC would dramatically decrease from the original 100 % to 57 % in the future RCP 8.5 scenario. The wetland SC would, therefore, be at higher risk of being unsaturated. The saturation probability of the wetland FL–WET would decrease from the historic 100 % to 63 % in the future RCP 8.5 scenario. The wetland AR is the only wetland that would remain 100 % saturated under all scenarios including RCP 8.5 scenario (Table 4).



## 4 Discussion

### 4.1 Difference and consistency of wetland hydrology models

Regarding the statistical results of the five models (Table 3), the relatively lower $R^2$ values of AR (ca. 21 %) compared with that of LP, are likely due to lateral water movement in AR with a coastal influence (*Johnston et al.*, 2005), which cannot be ignored but is generally hard to simulate. The $R^2$ values of FL wetland sites were lower (at percentages of 28 %, 43 %, 39 %) than that of the other three sites (AR, LP, and SC), likely due to the higher sensitivity

of the lower water table to the warming and strongly changing precipitation.

    Furthermore, the different regression coefficients of climatic-hydrological parameters (P-PET and antecedent water table at t-1 time step) and remarkably different intercepts (Table 3) among the five wetlands indicate different major controls for each of the wetland types. The trend that the model coefficients are similar between similar wetland types makes the regression model reasonable and acceptable. For example, the model shows much higher (approximately

ten times) intercepts in wetland LP (-19.55) and wetland FL–UP (-23.17), compared to wetland FL–WET (-1.36) and wetland SC (-3.79). This is reasonable, since both wetland FL–WET and wetland SC are depression wetlands, or geographically isolated wetlands (*Tiner et al.,* 2016) (i.e., ponds within flat landscape) surrounded by uplands, in which case local climatic-hydrological parameters would be the major controls and thus would have smaller intercepts. However, site FL–UP has sandy soils and found on higher elevation comparing to FL-WET on a flat landscape. The

artificial drainage systems for LP could be the control in addition to the climatic-hydrological parameters involved in the models. Hence, the much higher intercepts of site FL–UP and site LP reflect the topographic and drainage management controls for these two wetland types.

### 4.2 Different controls on future wetland hydrology in different wetlands

     Although the same model structure can be applied to all the five wetlands and had good performance, a closer

comparison shows different influence on the wetland hydrology. For example, the annual water tables in the future RCP 8.5 scenario decline the most in wetland FL. This may not only be due to the PET increase, which is similar to that of the other three sites (AR, LP, and SC), but also because the precipitation decreases at the same time in the wetland FL, while it increases at the other sites.

     Both precipitation and PET are predicted to be different between wetlands AR and LP. In the future RCP 8.5

scenario, the increment of PET in LP is slightly (8 %) higher than that in AR (Table 5), and the increment of precipitation in LP is 170 % that of AR. The change in P-PET, however, is generally similar between the two wetlands (-175 mm for LP and -184 mm for AR, Supplementary Table S1-S2). Despite the similar P-PET changes, the future water table changes in AR and LP are remarkably different. The annual water table of LP will decrease 19 cm compared to 3.75 cm for AR from the period of 1980-1999 to 2080–2099. The dramatic differences, which are

reflected by the different intercepts of the models, may be due to both management conditions in LP and the sea level rise effects in AR. Regarding the management conditions, wetland AR is undisturbed natural bottomland hardwoods while LP is highly managed pine plantation forests. LP has well-established ditches for drainage, with a flowline below the surface of the water table so that the hydraulic head of the drain is smaller than the hydraulic head of the

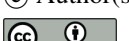

water table in the surrounding soil. The drainage ditch outflow on the watershed (pine plantation, NC) was closely
(e.g., $R^2$=0.75) related to the water table depth (*Amatya et al.*, 2006). Notably, regarding the AR wetland, the local
hydrologic drivers (not directly considered by the model, e.g., sea level rise) may increasingly impede the predicted
decreasing water table. Thus the sea level rise related hydrology may counter the predicted future water table decline.

Wetland type also contributes to the different water table dynamics. The water table changes differ significantly
from the baseline to the future RCP 8.5 scenario for FL–WET and FL–UP with the different topography condition.
The more significant change in FL–WET suggests that depression wetlands may be more sensitive to climate change
compared to uplands, consistent with the results of another study (*Lu et al.*, 2009).

Thus, the different responses of the future water table to climate change in wetlands with climatic and topographic
gradients and management conditions demonstrate the necessity and importance of developing wetland-specific
hydrologic models in specific regions.

### 4.3 Implications

#### 4.3.1 Efficient modeling of wetland water table dynamics

Compared to lumped (e.g., DRAINMOD–FOREST, Tian et al., 2015) or distributed parameter models (e.g., MIKE
SHE, Lu et al., 2009) the empirical hydrological models developed in this study is simple. However, our models well
explained the different water table dynamics for multiple wetland sites across the SE US region. Those differences in
wetland hydrological response to climate change imply different priority that wetland management strategies should
be different according to the site individual characteristics. For example, the differences between FL–UP and FL–
WET suggest that depressional wetlands have higher sensitivity to the climate change; the differences between AR
and LP suggest the importance of integrating the mechanisms of water table responds to sea level rise and extreme
storm events in AR site.

The empirical hydrological models performance well at the site and region levels, and can be empirically
incorporated into biogeochemical models, landscape and larger scale models. For example, the empirical hydrological
models can be linked with local soil respiration or regional CH4 emission models. Such empirical approach should be
compared to process-based physically based hydrological models to effectively model the biogeochemical change
under a changing climate.

#### 4.3.2 Biogeochemical cycles

Previous studies report that increases in temperature (3 °C–5 °C) and precipitation (146–192 mm/year) would lead
to a 175 % increase in the methane emissions (*Shindell et al.*, 2004). Wetland C emissions in forested wetlands could
be highly linked (e.g., a logarithmic relationship) to drought or flooded periods (*Moore and Knowles*, 1989). In AR,
*Miao et al.*, (2013) found that 93 % of the annual average soil $CO_2$ efflux of 960–1103 g C m$^{-2}$ in 2010 was released
in the nonflooded periods. Our study suggests that the non-flooded period would increase by 13 % (from 49 % to
62 %, assuming no rise in water table due to sea level rise) in the late 21$^{st}$ century. This means that the $CO_2$ efflux may
increase by 116–133 g C m$^{-2}$ accordingly. Other studies suggest that gross ecosystem productivity and the available
carbohydrate substrates for soil respiration would decrease with droughts (*Noormets et al.*, 2008). Wetland trees may





also alter their use and allocation of nutrients (e.g., N cycling) in response to the changing availability of water (Vose
et al., 2016).

### 4.3.3 Droughts and wildfires

The projected warming and long-term future drying indicate an increasing threat of droughts and wildfire in the
studying area (*Mitchell et al.*, 2014). Due to droughts, plant distributions may be shifted (*Desantis et al.*, 2007;
*Mulhouse et al.*, 2005), trees may become increasingly susceptible to attacks by pests and pathogens (*Schlesinger et
al.*, 2015). A warmer and longer growing seasons mean increased possibility of droughts and occurrence of wildland
fires (*Vose et al.*, 2016). Furthermore, increasingly frequent wildfire would release more carbon and stimulate more
greenhouse gas (GHG) emissions (e.g., CH4 production) with more biomass burning (*Medvedeff et al.*, 2015),  making
wetland forests a carbon sink rather than a source (*Westerling et al.*, 2006). Thus, the management challenges in
restoring wetland forests and reducing greenhouse gas emissions will substantially increase.

### 4.3.4 Wildlife and habitats

The predicted long-term drying (e.g., FL–WET) may greatly affect the biological diversity and metapopulations
of the studying wetlands by impacting the inter-wetland movements, recruitment, recolonization, and genetic
exchange of many species (*Moor et al.*, 2015; *Osland et al.*, 2013). Long-term drying could reduce the dispersion
among wetlands, and increase the isolation of primarily aquatic species such as cricket frogs (*Acrisgryllus*), pig frogs
(*L. gryllio*), swamp snakes (*Seminatrix pygaea*), and water fowl (*Davis et al.*, 2017; *Murphy et al.* 2016). The density
of waterfowl broods was found to be greater on impoundments than on seasonally flooded wetlands (*Connor and
Gabor*, 2006). Changes in the water table level of even less than 10 cm (predicted to decline from 7 cm to 28 cm
among the studied wetlands) may have profound effects on the habitat choice, species composition and provide
conditions which favour certain species or communities over those currently dominant in a given wetland (*Reddy and
DeLaune*, 2008). Brent geese would switch habitats within a water level span of 30 cm (*Clausen*, 2000). An equation
linking decay coefficient for a specific habitat type and the water table was even illustrated (*Bouma et al.*, 2014).
Temperature increase of 2 °C (projected to be 4 °C in this study) in Florida would influence co-occurring mangrove
and salt marsh plants (*Coldren et al.*, 2016). This supports the hypothesis of significantly far-reaching influences and
higher risks on wildlife habitats shifts for the studied wetlands in the SE US.

### 4.4 Uncertainty

Although the models developed by this study are efficient for simulating the historic and future water levels at
multiple wetlands, the models do not account the full physical processes that govern wetland hydrological cycles.
Thus, there are uncertainties on the hydrological response to extreme events such as droughts or floods that have not
occurred in the past. In addition, wetlands are not isolated, thus a landscape approach is needed to accurately model
wetland hydrology.

In addition to uncertainty of hydrological model structure, uncertainties associated with future climate change data
exist. GCMs projections are inherently inaccurate for small scale studies in spite of model bias corrections have been





implemented and multiple models are used in this application. Uncertainty in predicting precipitation is challenging in particular. Compared to previous studies using idealized climate data or climate data from a single GCMs, our approach assembling climate data from 20 GCMs and applying separate models to multiple wetlands  represents the best option to project hydrological response at the regional scale. Idealized or stochastically generated climate conditions were assumed in most previous models to illustrate impacts of climate change (*Chen et al.*, 2016).  Climate data from single (*Greenberg et al.*, 2015; *Wang et al.*, 2015) has been used in wetalnd hydrological resposne, but using several GCMs (*Chen et al.*, 2012; *Meinshausen et al.*, 2011) could result in more realistic results. However, different GCMs and future scenarios produce very different climate projections. The differences are even greater when applied to localized areas (*Alo and Wang et al.,* 2008). Multiple and overall GCM data can provide a better full-scale estimate (*Hessami et al.*, 2008).

## 5. Conclusions



The empirical hydrological models developed in this study are able to simulate water table dynamics for different types of wetlands across the southeastern U.S. With the antecedent water table, precipitation, and potential evapotranspiration as the main predictors of the water table at a 15-day time step, the developed models are simple but powerful to provide useful wetland hydrology information under a range of climatic and management conditions. Under climate change, the decrease in water availability is predicted to be a dominant factor for all five wetlands, resulting in a drier future in the study region, especially the late 21[st] century. This study confirms the hypothesis that climate change may have a significant but varying influence on the hydrology of different forested wetlands in the southeastern U.S.


This study may serve as a basis for future regional studies to understand the coupling between wetland hydrology and climate and quantify the role of wetlands in regulating water and energy balances, and climate feedbacks. Furthermore, given the close relationships between hydrology and biogeochemical cycles, vegetation distribution, fire regimes, and wildlife habitat, our study results suggest that the ecosystem functions of southern forested wetlands will be substantially impacted by future climate change. Climate change assessment on wetland forest ecosystems and adaptation management planning must first evaluate the impacts of climate change on wetland hydrology. Further studies are needed to explore the mechanisms of how climate change physically affects wetland water table dynamics and associated biogeochemical and ecological processes.






**Acknowledgments**

We would like to thank Heather Dinon Aldridge from NC State University for providing CMIP5 global climate model output. This study is supported by the National Science Foundation grant AGS-1147608, National Institute of Food and Agriculture, United State Department of Agriculture Grant 2014-67003-22068 and 2011-67009-20089, and the China Scholarship Council Fellowship 2015-0604-0157 to J. Zhu. Partial support is provided by the Eastern Forest Environmental Threat Assessment Center, USDA Forest Service.

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



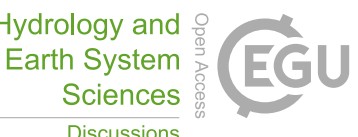

**Tables**

**Table 1 Characteristics of the studied wetlands.**

| Wetland | Coordinate | Climate (mean T and P) | Soil | Vegetation | wetland type | References |
|---|---|---|---|---|---|---|
| AR | 35°47' N, 75°54' W | 16.8 ℃ in July, 6.8 ℃ in Jan; 1298 mm (1971-2010) | Pungo (41 %) Longshoal (32 %) | Black gum, swamp tupelo, bald cypress (overstory); fetterbush, bitter gallberry, red bay (understory) | Natural wetlands; tree stand density of 2320 ± 800 stems/ha | (*Miao et al.*, 2013; *Moorhead and Brinson*, 1995) |
| LP | 35°48' N, 76°40' W | 26.6 ℃ in July, 6.4 ℃ in Jan; 1320±211 mm (1945-2008) | Belhaven Series, 20–95 % organic content in the top 50 cm and sandy loam underneath (*Diggs*, 2004) | Hardwoods, loblolly pine from 1992 | Artificially managed lower coastal plain forested wetland with tree density of 1660 trees/ha | (*Sun et al.*, 2010; *Tian et al.*, 2012; *Tian et al.*, 2015) |
| FL | 29°48' N, 82°24' W | 27℃ in July, 14 ℃ in Jan; 1330 mm | Top organic and sandy (Pomona fine sand) soil and underlying impermeable blue-greenclays | Flatwoods: Pond cypress, slash pine, swamp tupelo (wetland); slash pine (overstory in upland); saw palmetto, gallberry shrubs (understory in upland) | Cypress swamps and depression wetlands | (*Lu*, 2006; *Lu et al.*, 2009; *Sun et al.*, 1998; Sun et al., 2000) |
| SC | 33°06' N, 81°06' W | 26.5℃ in July, 12.5 ℃ in Jan; 1193 mm | Coxville series (fine, kaolinitic, thermic Typic Paleaquults). | Bottomland hardwoods (water oak, willow oak, swamp tupelo) | Carolina bay | (Dai et al., 2010; Dai et al., 2013; *Pyzoha et al.*, 2008; *Sun et al.*, 2006) |





**Table 2 Raw data summary.**

| Wetlands / Data types | | AR | LP | SC | FL − UP | FL − WET |
|---|---|---|---|---|---|---|
| Model development | Meteorological data | 07/02/2009–01/01/2011 | 01/01/2005–12/31/2012 | 01/01/1997–12/31/2002 | 01/01/1992–12/31/1996 | 01/01/1992–31/12/1996 |
| | Interval | 30 min | Daily, with some data missing | Daily | Daily | Daily |
| | Water table data | 03/19/2009–12/31/2011 | 01/01/2005–12/31/2012 | 01/01/1997–12/31/2002 | 01/01/1992–12/31/1996 | 01/01/1992–31/12/1996 |
| | Interval | Daily | Daily | Daily | Daily | Daily |
| Validation data | Model development Year | 2009–2010 | 2009–2012 | 1997–2000 | 1993–1994 | 1993–1994 |
| | Validation year | 2011 | 2005–2008 | 2001–2002 | 1992, 1995–1996 | 1992, 1995–1996 |
| | Interval | 15 days | 15 days | 15 days | 15 days | 15 days |
| Prediction data | Meteorological data | 01/01/1950–12/31/2099 | 01/01/1950–12/31/2099 | 01/01/1950–12/31/2099 | 01/01/1950–12/31/2099 | 01/01/1950–12/31/2099 |
| | Interval | 30 min | 30 min | 30 min | 30 min | 30 min |
| References | Data collection methods | *Miao et al., 2013* | *Noormets et al., 2010; Sun et al., 2010; Tian et al., 2015* | *Sun et al., 2006* | *Lu et al. 2009; Sun et al., 2000* | *Lu et al. 2009; Sun et al., 2000* |






**Table 3 Results for regressions of water table for five wetlands in the Southeastern United States.**

| wetland | $\alpha_{i0}$ | $\beta_{i1}$ | $\gamma_{i1}$ | $R^2$ | p |
|---------|------|------|------|------|------|
| AR (i=1) | -1.24 | 0.1137 | 0.7698 | 0.81 | <0.001 |
| LP (i=2) | -19.55 | 0.3750 | 0.8530 | 0.83 | <0.001 |
| FL–UP (i=3) | -23.17 | 0.3963 | 0.7206 | 0.69 | <0.001 |
| FL–WET (i=4) | -1.36 | 0.2360 | 0.8707 | 0.78 | <0.001 |
| SC (i=5) | -3.79 | 0.1450 | 0.82 | 0.71 | <0.001 |

Note: i is the number of the wetlands, i=1, 2, 3, 4, 5, t denoted the time periods, $\alpha_{i0}$ is the intercept, $\beta_{in}$ is the coefficient of the variable $X_n$ of the i wetland, $\gamma_{i1}$ is the coefficient of the antecedent water table at t-1 time step of the i wetland, $R^2$ denotes the coefficient of determination, and P donates the probability value when confidence level was at 95 % (Unites: mm/15 days).

**Table 4 Annual changes of variables from baseline scenario to scenario RCP 8.5 of five wetlands in the Southeastern United States.**

| Wetland | WT changes (cm) | Baseline annual WT (cm) | P (mm) | PET (mm) | P minus PET (mm) | AT (Deg C) |
|---------|------|------|------|------|------|------|
| AR | -4 | 0 | 37 | 221 | -184 | 3.9 |
| LP | -19 | -100 | 63 | 238 | -175 | 4.3 |
| FL–UP | -17 | -73 | -21 | 267 | -289 | 4.0 |
| FL–WET | -22 | 2 | -21 | 267 | -289 | 4.0 |
| SC | -7 | -16 | 60 | 266 | -207 | 4.3 |

Notes: WT is water table, P is precipitation, PET is potential evapotranspiration, AT is air temptation.





**Table 5 A summary of 15-day water table fluctuations in growing season under future RCP 8.5 scenario of five wetlands in the Southeastern United States**

| Wetlands | Lowest WT (cm) | PB of WT<0 cm | PR85 of WT<0 cm | PB of WT>-30 cm | PR85 of WT>-30 cm |
|---|---|---|---|---|---|
| AR | -17 | 49 % | 62 % | 100 % | 100 % |
| LP | -164 | 100 % | 100 % | 0% | 0 % |
| FL–UP | -124 | 100 % | 100 % | 0% | 0 % |
| FL–WET | -46 | 40 % | 93 % | 100 % | 63 % |
| SC | -42 | 100 % | 100 % | 100 % | 57 % |

Note: WT is water table, PB is the probability in baseline period, PR85 is the probability in RCP 8.5 period (2080-2099, future scenario F4). The water table >-30 cm is for growing season condition.

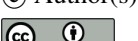



**Figures**

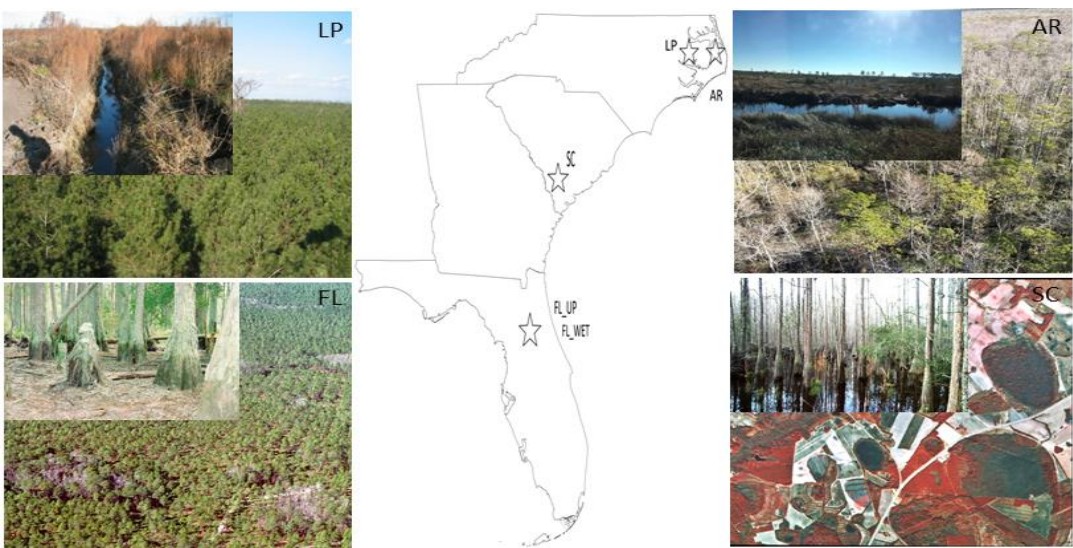

**Fig. 1 Study area, where the star symbol marks the study site location. Wetland AR: wetland of Alligator River National Wildlife Refuge in North Carolina; wetland LP: wetland of loblolly pine plantation in North Carolina; wetland SC: wetland in South Carolina; wetlands in Florida: wetland FL–UP (upland in Florida) and FL–WET (wetland in Florida).**







**Fig. 2 Comparison of observed and simulated 15-day water table depth in five wetlands in the Southeastern United States.**
**(a) site AR; (b) site LP; (c) site FL–UP; (d) site FL–WET; (e) site SC. WT is water table, Mea is the observed data, PRE is**
**the predicted data.**





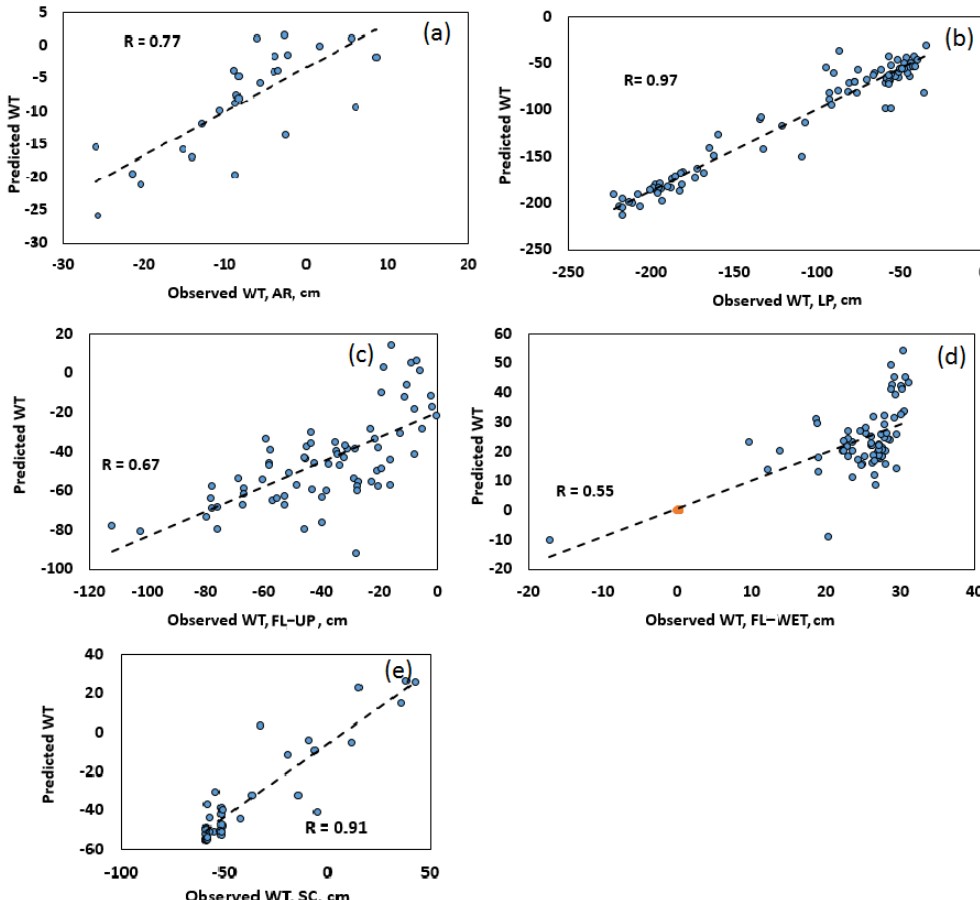

**Fig. 3 Scatter plots of the observed and predicted mean water table in five wetlands in the Southeastern United States. (a) AR; (b) LP; (c) FL–UP; (d) FL–WET; (e) SC; Dashed lines are 1:1 line.**



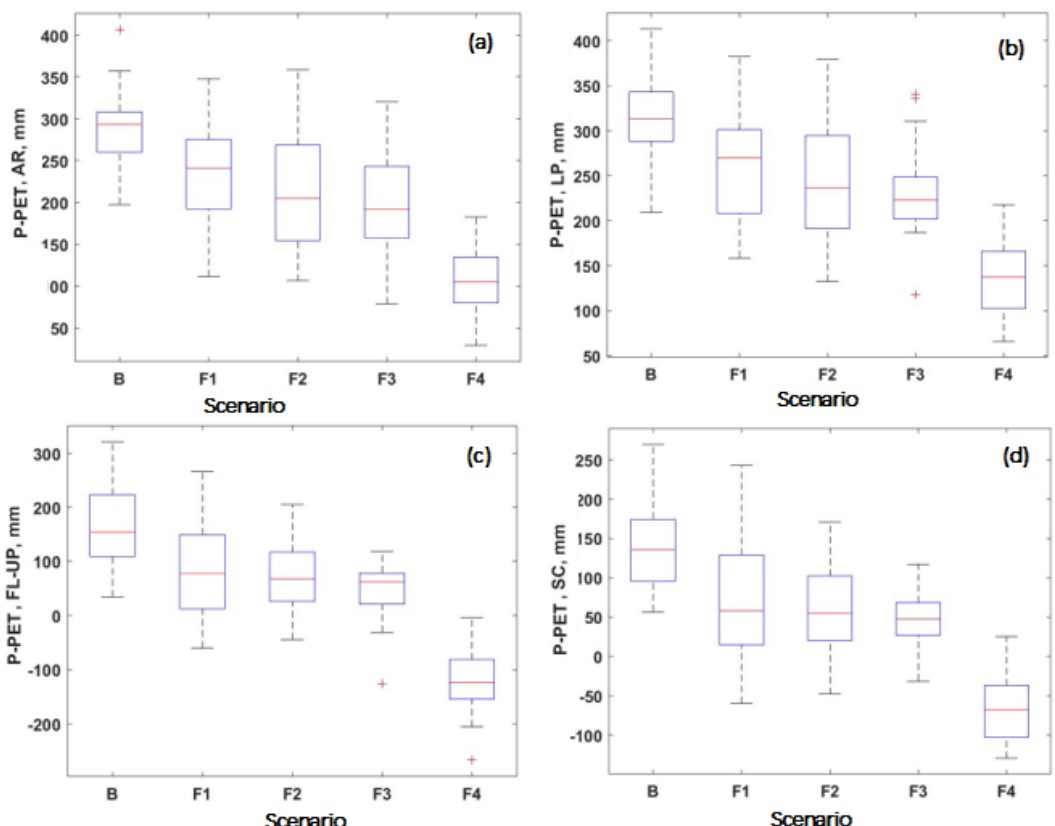

**Fig. 4 Total annual precipitation minus potential evapotranspiration of 20 GCMs in five wetlands in the Southeastern United States (unit: mm), (a): AR, (b): LP, (c): FL, (d): SC.** B:1980–1999, historical baseline period; F1:2040–2059, RCP 4.5, future scenario 1; F2:2080–2099, RCP 4.5, future scenario 2; F3:2040–2059, RCP 8.5, future scenario 3; F4:2080–2099, RCP 8.5, future scenario 4;





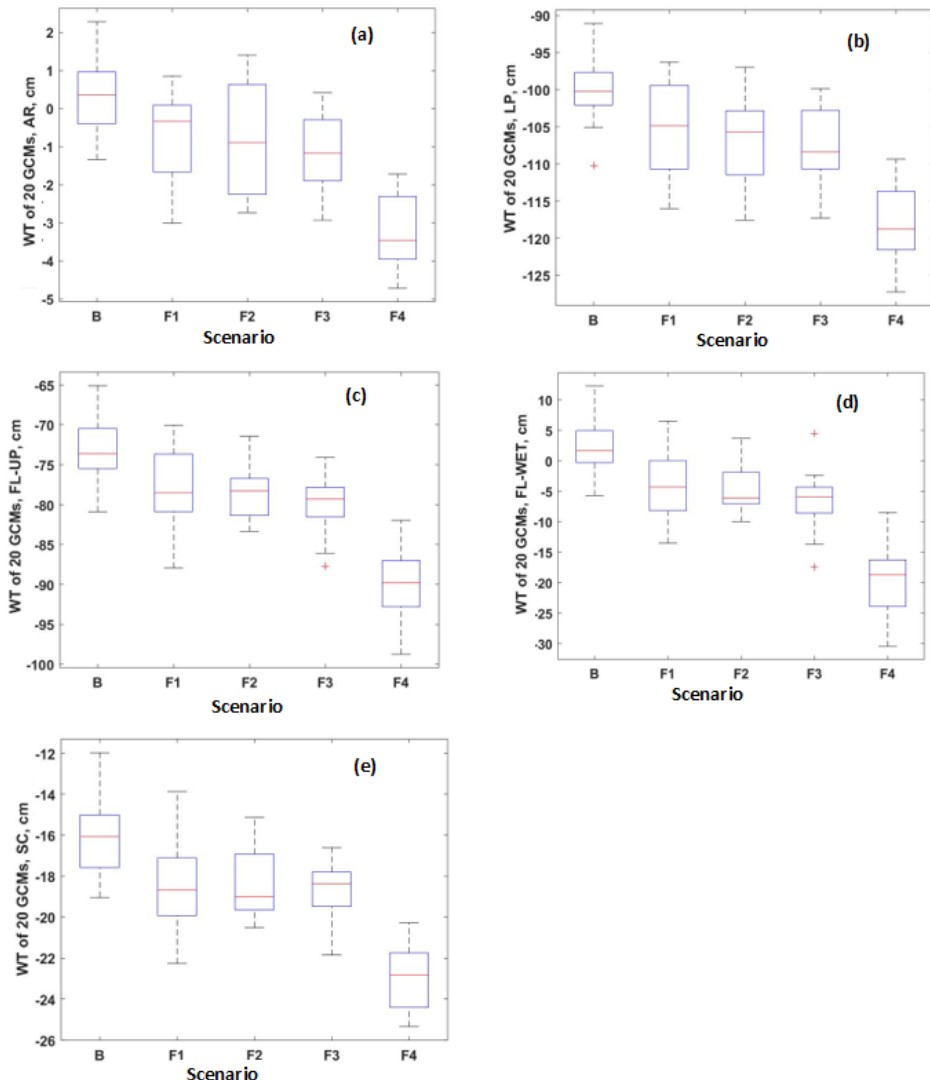

**Fig. 5 Mean predicted annual water table of 20 GCMs in five wetlands in the Southeastern United States (unit: cm), (a) AR,**

**(b) LP, (c) FL–UP, (d) FL–WET, (e) SC.**

Note: B:1980–1999, historical baseline period;

F1:2040–2059, RCP 4.5, future scenario 1; F2:2080–2099, RCP 4.5, future scenario 2;

F3:2040–2059, RCP 8.5, future scenario 3; F4:2080–2099, RCP 8.5, future scenario 4;





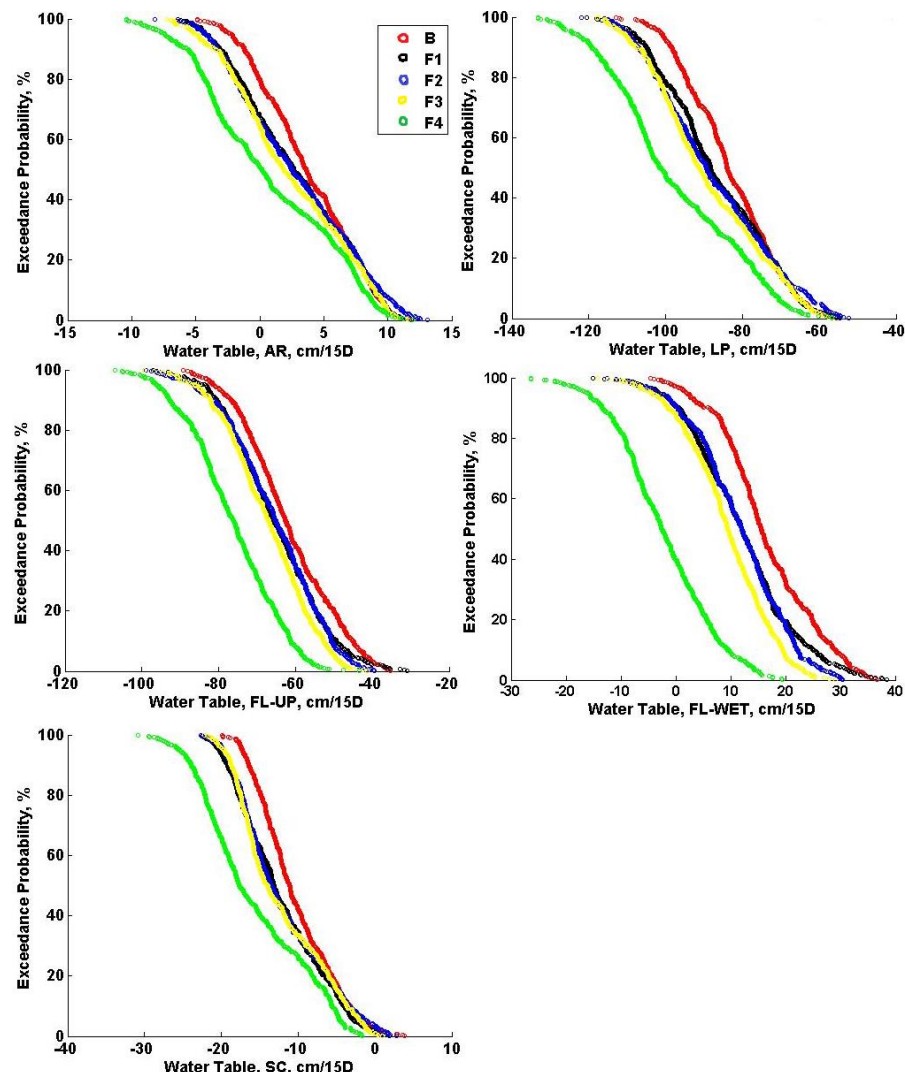

**Fig. 6 Exceedance probability of the mean predicted water table in the growing season of 20 GCMs in five wetlands in the Southeastern United States (unit: cm/15 days), (a) AR, (b) LP, (c) FL–UP, (d) FL–WET, (e) SC, respectively.** B:1980–1999, historical baseline period; F1:2040–2059, RCP 4.5, future scenario 1; F2:2080–2099, RCP 4.5, future scenario 2; F3:2040–2059, RCP 8.5, future scenario 3; F4:2080–2099, RCP 8.5, future scenario 4.