# Peer review of "Modeling the Potential Impacts of Climate Change on the Water Table Level of Selected Forested Wetlands in the Southeastern United States"

_Hydrology and Earth System Sciences, 2017_

## Referee Comment (RC1) · Anonymous Referee #1 · 17 May 2017

This study uses a linear regression model to estimate future water table changes in five forest sites in Southeastern US. The topic is interesting. However, this study lacks of innovation in climate change impact assessment. Besides the simple regression model, there are critical issues/errors in the methodology (see major comments). Please find my detailed comments below.

Major Comments

1. In fitting the regression model, water table at the current time step is the dependent variable while water table at the previous time step is included as one of the independent variables. This is not reasonable given the potential autocorrelations between current water table and antecedent water table especially at the daily scale. In fact,

the unbelievably high R2 value of 0.97 in predicting water table at the LP site could be due to the inclusion of antecedent water table in the statistical model. What's more, how can future water table be predicted by the statistical model which requires inputs of antecedent water table conditions?

2. There are major issues related to the short calibration and validation periods for the statistical model. For example, two years of data is used for fitting regression model for the AR site while one year is used for validation. I am wondering whether climatic conditions in the validation year is significant different from the calibration year? Future climate especially for the later periods of 21st century would be quite different from the calibration periods based on which the regression model is constructed. Therefore, the historical relations trained from such a short time period may not hold in the future with significant changes in climate.

3. The downscaled GCM climate should be validated for the baseline period in the study sites before it can be used for future predictions.

Minor Comments

1. In Section 80, RCP stands for "Representative Concentration Pathway" rather than "Regional Concentration Pathways"

2. Hamon's equation is selected for estimating PET. Justifications on this should be added.

3. In section 130, the estimated PET is adjusted to match "realistic" PET values for forests. What are the realistic PET values for forests?

4. The climate for the baseline period is based on observations or GCM simulations?

5. A table with a brief description of the GCMs should be added.

6. This study focus on the projection of water table depth at five forest sites. The title should mention "water table depth" rather than "hydrology" which is a broad concept.

---

## Referee Comment (RC2) · Anonymous Referee #2 · 17 May 2017

General Comments

The discussion paper uses an empirical approach to determine hydrological effects on climate change for 5 wetland types in the southeastern U.S. The paper generally has scientific significance in that it tries to address uncertainties associated with climate change, and the overall structure of the paper is clear and concise. However, the paper lacks rigorous evaluation of the model and results. The general model structure appears flawed (see comments below), and model results are seemingly taken at face value. For example, the authors do not address a major source of uncertainty associated with climate change: water use efficiency (WUE). Climate change is associated with increases in $CO_2$, not just temperature, and increased $CO_2$ is known

to increase WUE, which would have major implications for the results presented here. There was also little consideration given to changes in water availability for vegetation, which drives actual ET. Additionally, the graphics and tables were lacking in quality and readability, and should be revised. There were many typographical and grammatical errors. Overall, this paper needs major revisions.

Scientific Comments

Line 78-79: Why is Greenberg et al. (2015) referenced here without discussing how it "satisfactorily" used an empirical model? All you say is that they used one.

Line 158-159: It is not clear what the rationale is for using lagged water table as an independent variable? It seems clear that the most recent water table value will be highly correlated to the water table now. Is this just using autocorrelation as a covariate? Consider revision.

Line 160: It seems like an autocorrelation covariance structure should be used given the time-series nature of the data.

Line 176: Water loss is also controlled by net groundwater flow, but more importantly by vegetation access to water and vegetation water use efficiency (WUE), which are not accounted for in the model. And because we know that WUE is strongly influenced by $CO_2$ concentrations, this appears to be a major deficiency in the model.

Line 186-195: Did you also test for assumptions of normality of residuals and homoscedasticity of residuals? If you did not take into account autocorrelation of covariance it is likely that these assumptions may be violated.

Lines 196-199: What did you find with Durbin's h? Did it support autocorrelation or not?

Line 234-236: Are these estimates for changes to PET based purely on temperature changes? This seems important to note.

Line 270-271: This sentence is the opposite of what is suggested by the figure and is confusing to interpret.

Lines 272-279: This section is very difficult to understand, especially trying to reconcile with figures. Suggest re-writing.

Lines 283-285: Where are the R2 values coming from? Are these ratios of R2 to other sites? Clarification needed.

Line 288: Did you statistically test that the model coefficients were similar? They do not seem too similar to me...

Where is the discussion of how the model did not perform well? The model appears to be much flashier and tends to overpredict relative to observed data? RMSE or some other metric would be useful as a comparison.

Technical Corrections

Line 56: "...and more powerful hurricanes landfall." Word choice here is awkward.

Line 58: "process-based study" should be "process-based studies"

Line 70: add "and" before "...their potential uses..."

Line 73-75: This sentence needs revision for clarity and grammar.

Line 75: "Performance such type of models..." a word is missing.

Line 84: change "increased" to "subsequent increases"

Line 88: There is an extra "s" after the parentheses

Line 289: change "higher" to "lower"

Line 387: Missing a word in "Climate change from single has been used..." and "we-talnd" is misspelled.

Line 625: Table 1 should have consistent formatting for each of the data in columns

for ease of comparison. Consider a more generic description of soils instead of series names.

Line 670: Figure 3(d) what is meant by the orange dots?

Line 680 and 685: Figures 4 and 5 are begging to have significance letters attributed to each boxplot.

Line 685: Figure 6 – These axes should be flipped for ease of interpretation. Also fix the legend so it doesn't look like it was drawn by hand. Consider changing the x-axis label and putting the site name in the figure panel itself.

---

## Author Comment (AC2) · 12 Jul 2017

**GENERAL COMMENTS:** The discussion paper uses an empirical approach to determine hydrological effects on climate change for 5 wetland types in the southeastern U.S. The paper generally has scientific significance in that it tries to address uncertainties associated with climate change, and the overall structure of the paper is clear and concise. However, the paper lacks rigorous evaluation of the model and results. The general model structure appears flawed (see comments below), and model results are seemingly taken at face value. For example, the authors do not address a major source of uncertainty associated with climate change: water use efficiency (WUE). Climate change is associated with increases in CO2, not just temperature, and increased CO2 is known to increase WUE, which would have major implications for the results presented here. There was also little consideration given to changes in water availability for vegetation, which drives actual ET. Additionally, the graphics and tables were lacking in quality and readability, and should be revised. There were many typographical and grammatical errors. Overall, this paper needs major revisions.

**RESPONSE:** We are very thankful for the reviewer's detailed reviews about the uncertainty associated with our modeling results related to other factors of future climate change, e.g. the increased WUE because of the increased $CO_2$, and the changes in water availability for vegetation. This work focuses on the wetland groundwater variability due to climate drivers' change such as precipitation and air temperature. We did not specifically consider the effects of increased $CO_2$ on vegetation growth and productivity which may further affect wetland hydrology in the study due to the following reasons.

First, some of the GCM models used here already contain a dynamics vegetation model (e.g., Yu et al., 2016); thus, vegetation responses to increased $CO_2$ have been considered in these GCMs. Second, previous results suggest that precipitation and surface air temperature are the two first order climate variables to drive wetland groundwater (Liljedahl et al., 2011). Third, quantitative analysis about the wetland evaporation/evapotranspiration comparison found that transpiration from canopy provided few contributions to evapotranspiration (Li et al., 2009).

We thank the reviewer to get our attention and will study the aspect in our future work although this is out of the scope of work of the present study. According to the comment, we added recommendations for future work in line 200-203 (in the revised manuscript, similarly hereinafter). The related references were added in the revised version.

"Li, Y. J., Zhou, L., Xu, Z. Z., and Zhou, G. S.: Comparison of water vapour, heat and energy exchanges over agricultural and wetland ecosystems, Hydrological Processes, 23, 2069-2080, 2009. (Line 571-573)

Liljedahl, A. K., Hinzman, L. D., Harazono, Y., Zona, D., Tweedie, C. E., Hollister, R. D., Engstrom, R., and Oechel, W. C.: Nonlinear controls on evapotranspiration in arctic coastal wetlands, Biogeosciences, 8, 3375-3389, 2011. (Line 574-576)

Yu, M., G. Wang, and H. Chen, Quantifying the impacts of land surface schemes and dynamic vegetation on the model dependency of projected changes in surface energy and water budgets, J. Adv. Model. Earth Syst., 8, 370–386, 2016. (Line 699-701)"

We have also improved the methodology (line 131-134, line 141-144, line 148-156, line 161-175, line 195-198, line 223-232, etc.), results (line 248-249, line 249-250, line 260-270, line 312-314, line 315-323, etc.), and discussion (line 326-332, line 439-440, etc.), re-plotted the graphs (line 725, line 730, line 740, line 745, line 755, and line 760) using high resolution and re-done the tables (line 700 and line 710) to improve the quality and readability as suggested.

**SCIENTIFIC COMMENTS:**

For the reviewer's convenience during re-review, we numbered his/her comments and included our corresponding responses, below.

**Point #1**

**COMMENT:** Line 78-79: Why is Greenberg et al. (2015) referenced here without discussing how it "satisfactorily" used an empirical model? All you say is that they used one.

**RESPONSE:** We much appreciate the reviewer's careful review. We further discussed the references in line 75-77 as follows:
"Greenberg et al. (2015) developed an empirical model and demonstrated its utility for climate-change planning by successfully forecasting the weekly hydrologic regimes (2012-2060) and examining the indirect impacts of climate change on biological diversity."

**Point #2**

**COMMENT**: Line 158-159: It is not clear what the rationale is for using lagged water table as an independent variable? It seems clear that the most recent water table value will be highly correlated to the water table now. Is this just using autocorrelation as a covariate? Consider revision.

**RESPONSE:** The lagged water table was actually considered as a covariate. The rationale to use the lagged water table was to account for its contribution to the current water table, in addition to the role of current climate and water availability, based on the considerations as follows:

   (i) This study adopted the well-established methodology of dynamic panel model widely used in statistics and econometrics. The dynamic panel modeling includes the first lag dependent variable coupled with the explanatory variables (e.g. P, PET, in this study). The model structure with a given lag effect were successfully used in the previous studies for hydroregime prediction (Greenberg et al., 2015, Webb et al., 2003), urban water demand prediction (Almendarez-Hernández et al., 2016; Arbués et al., 2004; Arbues et al., 2010, Lyman, 1992), and energy-food-water interaction modelling (Liu et al., 2017; Ozturk, 2015). Lyman's (1992) and Ozturk (2015) confirmed the adjustments significance of minimizing heterogeneity in the traditional Ordinary Least Squares assumptions by including the first lagged dependent variable. Webb et al. (2003) improved the sensitivity and explanatory power of the hourly based water-air temperature regression models by incorporating a lagged response of water temperature. In a wetland

hydrology and climate change study, Greenberg et al. (2015) successfully forecasting the hydroregimes of multiple wetlands by modeling the water table depth using water level of the prior week and precipitation as predictors of current water table.

(ii) The statistical model structure has physical meaning and can be viewed from the perspective of water balance. A lagged effect of water table was supported by the water balances and the water table dynamics of wetlands. It is due to the fact that wetland groundwater has memories which can be carried beyond the next season as to influence the water balance in the coming years (Miguez-Macho and Fan 2012). Firstly, the water balance of the five selected wetlands can be written as $WT_t - WT_{t-1} = \alpha + \beta(P_t - ET_t) + \varepsilon_t$. It indicates the causality balances between water tables changes (left side terms of the above equation i.e., $WT_t - WT_{t-1}$) and its water availability changes (right side terms of the above equation). The item of $WT_{t-1}$ depending on the time scale can be considered as the memory effect of water tables. By moving the memory item ($WT_{t-1}$) to the right side, the different statistical coefficient of $WT_{t-1}$ can reflect the different memory characters/effects in the five selected wetlands. Based on these reasons, we believe the model structure that includes a lagged water table in this study can offer more information concerning not only water tables changes due to changes in climate variables, but also the different memory effects of different wetlands in this region.

(iii) All the information is not contained in the antecedent water table conditions of wetlands. The lagged water table only offers the basis for the current condition, however, the forcings (e.g. climate and water availability in this study) alters the water table depth. In another word, the water table depth would always decrease/increase along with a given initial discharge/recharge condition for the wetland. But, a wetland actually alternatively discharges or recharges for the flatness, thus water tables fluctuated with the forcings. Therefore, both the forcings and the lagged water table would determine the water tables for a wetland. What's more, when using the water table at LP site with forcings from FL-WET site, the statistics show that in spite of the same antecedent condition, the $R^2$ becomes poorer to 0.54 from 0.83.

(ⅳ) From the perspective of the proved wide-sense stationary first-order autoregressive process of water tables in the five selected wetlands, the variance will not change with the autoregressive process introduced into the statistic model. For an autoregressive process given by: $Y_t = \alpha + \beta Y_{t-1} + \varepsilon_t$, where $\varepsilon_t$ is a white noise process

with zero mean and constant variance $\sigma_\varepsilon^2$. The first-order autoregressive process is wide-sense stationary when and only when $|\beta|<1$, because it is the output of a stable filter with a white noise input (Mills, 1990). In the wetlands analyzed in the study, the coefficient of $WT_{t-1}$ are $<1$ (Table 4), indicating a wide-sense stationary autoregressive process of water table. Thus the variance of the process does not change with simulation over time, and the intertemporal effect ($\beta^n \varepsilon_1$) of shocks diminishes toward zero in the limit.

(ⅴ) From the perspective of the independence of the explanatory variables, introduce of antecedent water table does not violate the independence requirements among the explanatory variables. Independence between the explanatory variables was satisfied since correlation coefficient between P-PET and $WT_{t-1}$ was very poor ($<2.7$) for the five selected wetlands in the study.

According to the comment, the justification was further concisely added in line 161-175 in the revised manuscript to make it clearer for readers. Also, the related new references were added to the revised manuscript as follows:

"Almendarez-Hernández, M., Avilés Polanco, G., Hernández Trejo, V., Ortega-Rubio, A., and Beltrán Morales, L.: Residential Water Demand in a Mexican Biosphere Reserve: Evidence of the Effects of Perceived Price, Water, 8, 428, 2016. (Line 467-469)

Arbués, F., Barberán, R., and Villanúa, I.: Price impact on urban residential water demand: A dynamic panel data approach, Water Resour Res, 40, 2004. (Line 475-476)

Arbues, F., Garcia-Valinas, M. A., and Villanua, I.: Urban Water Demand for Service and Industrial Use: The Case of Zaragoza, Water Resour Manag, 24, 4033-4048, 2010. (Line 477-478)

Liu, G., Yang, Z., Tang, Y., and Ulgiati, S.: Spatial correlation model of economy-energy-pollution interactions: The role of river water as a link between production sites and urban areas, Renewable and Sustainable Energy Reviews, 69, 1018-1028, 2017. (Line 574-576)

Lyman, R. A.: Peak and off-peak residential water demand, Water Resour Res, 28, 2159-2167, 1992. (Line 586)

Mills, Terence C. Time Series Techniques for Economists. Cambridge University Press, 1990. (Line 600)

Ozturk, I.: Sustainability in the food-energy-water nexus: Evidence from BRICS (Brazil, the Russian Federation, India, China, and South Africa) countries, Energy, 93, 999-1010, 2015. (Line 627-628)

Webb, B., Clack, P., and Walling, D.: Water–air temperature relationships in a Devon river system and the role of flow, Hydrological processes, 17, 3069-3084, 2003. " (Line 688-689)

**Point #3**

**COMMENT**: Line 160: It seems like an autocorrelation covariance structure should be used given the time-series nature of the data.

**RESPONSE:** We view that "an autocorrelation covariance structure" here as "covariance structure of autocorrelation". For an autoregressive process given by: $Y_t = \alpha + \beta Y_{t-1} + \varepsilon_t$, where $\varepsilon_t$ is a white noise process with zero mean and constant variance $\sigma_\varepsilon^2$. The autocovariance is $B_n = E(X_{t+n}X_t) - \mu^2 = \frac{\sigma_\varepsilon^2}{1-\beta^2}\beta^{|n|}$, where $\mu$ is the model mean, and n is the time step (Mills, 1990). In this study, autoregressive process is wide-sense stationary ($|\beta|<1$), thus $\frac{\sigma_\varepsilon^2}{1-\beta^2}\beta^{|n|}$ diminishes toward zero in the limit in the general model form (Line 180). Besides, the autocorrelation nature for the given data time-series was first tested to select the final variables. The autocorrelation of water tables with lag time of 0 day, 15 days, 30 days, 45 days, and 60 days were tested, as well as their correlations with the other explanatory variables. Consequently, in the final model (line 246), only the 15-day (one time step) lagged water table was chose as an explanatory variable for the best statistical results. Thus only a constant variance $\sigma_\varepsilon^2$ exists in the final selected wide-sense stationary first order autoregressive process.

**Point #4**

**COMMENT**: Line 176: Water loss is also controlled by net groundwater flow, but more importantly by vegetation access to water and vegetation water use efficiency (WUE), which are not accounted for in the model. And because we know that WUE is strongly influenced by CO2 concentrations, this appears to be a major deficiency in the model.

**RESPONSE:** Please see our replies to the General Comment (page 1-2). We agree with reviewer on the hydrological processes identified. However, given the empirical nature of our model, these factors are not explicitly considered. We have also added some discussion regarding the potential uncertainty originating from discounting the effects of increase in $CO_2$ on WUE in the revised version (see lines 200-203):
"The change in atmospheric $CO_2$ is likely to affect water use by trees through altering plant water use efficiency (WUE) (Brummer et al., 2012), but this process was not considered in this study. In addition, lateral water loss/gain from net groundwater flow was not simulated explicitly."

The related new reference was added to the revised manuscript as follows:

"Brummer, C., Black, T. A., Jassal, R. S., Grant, N. J., Spittlehouse, D. L., Chen, B., Nesic, Z., Amiro, B. D., Arain, M. A., Barr, A. G., Bourque, C. P. A., Coursolle, C., Dunn, A. L., Flanagan, L. B., Humphreys, E. R., Lafleur, P. M., Margolis, H. A., McCaughey, J. H., and Wofsy, S. C.: How climate and vegetation type influence evapotranspiration and water use efficiency in Canadian forest, peatland and grassland ecosystems, Agricultural and Forest Meteorology, 153, 14-30, 2012." (Line 490-494)

**Point #5**

**COMMENT**: Line 186-195: Did you also test for assumptions of normality of residuals and homoscedasticity of residuals? If you did not take into account autocorrelation of covariance it is likely that these assumptions may be violated.

**RESPONSE:** The normality and the homoscedasticity for both the five sites were tested before the models were applied to the prediction. The residual plots of the five specific models showed that errors are homoscedastic, and both the residuals and the normal probability plot showed the normality of the residuals in the five selected wetlands. With introducing the autoregressive variable, the Durbin's h also indicated the autocorrelation disturbance process. We also added the clarification to the revised version in line 248-249:

"The residual plots and the normal probability plot of residuals showed the normality and homoscedasticity of residuals of the five specific models." The related figures of AR site were shown as an example as follows:

[Figure]

[Figure]

[Figure]

Figure 1 Residuals and normal probability plots of AR site, a) P-PET residual plot, b) WTt-1 residual plot, c) normal probability plot. The residuals and standard residuals of the observed Y (water table) have a mean of zero ($6\times10^{-16}$, $1\times10^{-16}$, respectively).

**Point #6**

**COMMENT**: Lines 196-199: What did you find with Durbin's h? Did it support autocorrelation or not?

**RESPONSE:** The Durbin's h tests for the models showed that all the five wetlands regression models support the autocorrelation, by comparing the lower and upper critical value from the Durbin-Watson Table. The results of DW test were added to the Table 4 in line 711 in the revised manuscript. We also added the statement to the revised manuscript in line 249-250 as follows:
"Durbin's h statistic showed that the five wetlands regressions support the autocorrelation disturbance process."

**Point #7**

**COMMENT**: Line 234-236: Are these estimates for changes to PET based purely on temperature changes? This seems important to note.

**RESPONSE:** Yes, PET is estimated based on air temperature only in addition to day length following Sun et al., (2002). The associated note was added in line 195-198 as follows:
"PET is mainly controlled by net radiation, air temperature, wind speed, air humidity (Hargreaves and Samani, 1982). Due to data availability, here we used the air temperature-based Hamon equation to calculate PET (Hamon, 1963) using widely

available temperature data. This Hamon's PET method has been widely used worldwide to estimate potential water uses, especially in this region (Sun et al., 2002)"

The new reference was added to the revised manuscript as follows:
"Hargreaves, G. H. and Samani, Z. A.: Estimating potential evapotranspiration, Journal of the Irrigation and Drainage Division, 108, 225-230, 1982." (Line545-546)

**Point #8**

**COMMENT**: Line 270-271: This sentence is the opposite of what is suggested by the figure and is confusing to interpret.

**RESPONSE:** The sentence was revised in line 312-314 as follows;
"In contrast to wetlands of LP, FL–UP, and SC, the wetlands AR and FL–WET show a lower probability (40 % for FL–WET, 49 % for AR) being ineffective to store surface water as a wetland in the baseline, but still significantly increasing to 62 % and 93%, respectively."
We also updated the table order in line 312 from (Table 4, Fig. 6) to (Table 6, Fig. 6).

**Point #9**

**COMMENT**: Lines 272-279: This section is very difficult to understand, especially trying to reconcile with figures. Suggest re-writing.

**RESPONSE:**  The section has been re-written (Line 315-323) as follows:
   "While LP, FL–UP and SC were all predicted to be ineffective to store surface water (water table < 0 cm) over the study period, the soil saturation status (water table depth still within 30 cm) were different (Table 6). Site LP and FL–UP would completely dry up from baseline to2099 based on the RCP 8.5 scenario. Wetland SC, which was saturated 100% of the time during baseline period, would also suffer significant dryness with saturation time period decreasing to 57 % in 2099. The wetland SC would, therefore, be at high risk of being unsaturated. The wetland FL-WET, however, would be the most sensitive one among the five with the most change of probability being ineffective to store surface water (increasing from 40% to 93% from the baseline period to 2099) and being saturated (decreasing from 100% to 63% ). Notably, the wetland AR would be the

only wetland that would remain 100% saturated under all scenarios including RCP 8.5 scenario (Table 6, Fig. 6)."

**Point #10**

**COMMENT**: Lines 283-285: Where are the R2 values coming from? Are these ratios of R2 to other sites? Clarification needed.

**RESPONSE:** The $R^2$ values are the coefficient of determination of the regressions model results. The clarification was added in the revised version in line 326-332 as follows: "the relatively lower $R^2$ values of AR (0.81) compared with that of LP (0.83), are likely due to lateral water movement in AR due to coastal influence (Johnston et al., 2005), which cannot be ignored but is generally hard to simulate. The $R^2$ values of FL-WET and SC sites (0.78 and 0.72, respectively) were lower than that of the North Carolina sites (AR and LP).It was likely due to the higher sensitivity of the wetland type (FL-WET site as a depression wetland, SC site as a Carolina bay, Table 1) to the warming and strongly changing precipitation. The lowest $R^2$ values lies in the FL-UP site (0.69) mainly for the uncertain contribution of the artificial managed drainage system."

**Point #11**

**COMMENT**: Line 288: Did you statistically test that the model coefficients were similar? They do not seem too similar to me…

**RESPONSE:** We much appreciate the reviewer's careful review. The sentence was deleted in the revised version.

**Point #12**

**COMMENT**: Where is the discussion of how the model did not perform well? The model appears to be much flashier and tends to overpredict relative to observed data? RMSE or some other metric would be useful as a comparison.

**RESPONSE:** We have addressed reviewer's concerns by adding RMSE values. The discussion was added in line 260-270 and line 439-440 as follows:

"Compared to observation years, the average water table was over-predicted by 1.4 cm for LP (-106.25 cm for observation, -104.85 cm for prediction, with root mean square error (RMSE) of 4.92 cm, similarly hereinafter), 1.97 cm for FL-WET (19.02 cm, 19.97 cm, with RMSE of 9.23 cm), and 1.3 cm for SC (-19.1 cm, -17.8 cm, with RMSE of 5.16 cm). Also, it was under-predicted at 2.11 cm for FL-UP (-48.97 cm, -51.08 cm, with RMSE of 5.9 cm), and 0.38 cm for AR (-4.19 cm, -4.57 cm, with RMSE of 3.71 cm). The under/over prediction may be explained by the different model capability for the different characters of the wetlands types (Cypress Ponds/Swamps, Carolina Bays, Pine Flatwoods, and Wet Pine, and natural Bottomland Hardwoods ecosystems). For example, for the FL-WET Cypress Ponds/Swamps, the water tables were relatively over-predicted during the normal period while the observations and the predictions matched better during the extreme dry year in 1993. It may be because of the higher water table sensitivity to the forcings and the sharper water table changes in a short term (two weeks) in FL-WET as a depression wetland. It also explained the good capability of the empirical models in the annual-scale water table averages, even in the sensitive FL-WET site. Overall, …" (line 260-270).

"Besides, limited observation data availability can contribute to model deficiency and uncertainty as well." (Line 439-440).

**TECHNICAL CORRECTIONS:**

**Point #1**

**COMMENT**: Line 56: "… and more powerful hurricanes landfall." Word choice here is awkward.

**RESPONSE:** In line 54-55, the phrase was revised to "more frequent and intense Atlantic hurricanes".

**Point #2**

**COMMENT**: Line 58: "process-based study" should be "process-based studies".

**RESPONSE:** The phrase was revised to "process-based studies" in line 56 in the updated version.

**Point #3**

**COMMENT**: Line 70: add "and" before "…their potential uses…"

**RESPONSE:** The "and" was accordingly added in line 67.

**Point #4**

**COMMENT**: Line 73-75: This sentence needs revision for clarity and grammar.

**RESPONSE:** The sentence was revised in line 70-71 as follows:
"In contrary, when applied over multiple sites, statistical models have advantages of both high efficiency and acceptable performance."

**Point #5**

**COMMENT**: Line 75: "Performance such type of models…" a word is missing.

**RESPONSE:** The sentence was revised as "Especially, performance of such empirical models…" in line 71-72.

**Point #6**

**COMMENT**: Line 84: change "increased" to "subsequent increases".

**RESPONSE:** The word was changed accordingly in line 82.

**Point #7**

**COMMENT**: Line 88: There is an extra "s" after the parentheses.

**RESPONSE:** The extra "s" was deleted in the revised version in line 88.

**Point #8**

**COMMENT**: Line 289: change "higher" to "lower".

**RESPONSE:** The word was accordingly changed to "lower" in line 335.

**Point #9**

**COMMENT**: Line 387: Missing a word in "Climate change from single has been used…" and "wetalnd" is misspelled.

**RESPONSE:** The sentence was revised as "Climate data from single GCMs (Greenberg et al., 2015; Wang et al., 2015) has been used in wetland hydrological response, …" in line 4235-436.

**Point #10**

**COMMENT**: Line 625: Table 1 should have consistent formatting for each of the data in columns for ease of comparison. Consider a more generic description of soils instead of series names.

**RESPONSE:** The Table 1 was reformatted and revised in line 700, and a more generic description of soils instead of series names were used for the sites. The climate data from different time series including the most of the observation years is to better reflect different climate background during model development.

**Point #11**

**COMMENT**: Line 670: Figure 3(d) what is meant by the orange dots?

**RESPONSE:** Figure 3 (line 740, as follows) was updated in the revised version with mistake fixed and the site names put in the figure panel itself. We also improved the quality of Figure 1 (line 725, as follows) and Figure 2 (line 730, as follows).

[Figure]

Fig. 3 Scatter plots of the observed and predicted mean water table in five wetlands in the Southeastern United States (unit: cm), Dashed lines are 1:1 line.

[Figure]

Fig. 1 Study area, where the star symbol marks the study site location. Wetland AR: wetland of Alligator River National Wildlife Refuge in North Carolina; wetland LP: wetland of loblolly pine plantation in North Carolina; wetland SC: wetland in South Carolina; wetlands in Florida: wetland FL–UP (upland in Florida) and FL–WET.

[Figure]

Fig. 2 Comparison of observed and simulated15-day water table in five wetlands in the Southeastern United States, WT is water table.

**Point #12**

**COMMENT**: Line 680 and 685: Figures 4 and 5 are begging to have significance letters attributed to each boxplot.

**RESPONSE:** We much appreciate the reviewer's careful review. The Figures 4 and 5 were revised in line 745 (as follows) and 755 with significance letters, site names in the panel itself, and the quality was improved as well.

[Figure]

Fig. 4 Total annual precipitation minus potential evapotranspiration of 20 GCMs in five wetlands in the Southeastern United States (unit: mm).

Note: Bseline:1980–1999, historical baseline period;

Mid 20-4.5:2040–2059, RCP 4.5; Late 20-4.5:2080–2099, RCP 4.5;

Mid 21-8.5:2040–2059, RCP 8.5, Late 21-8.5:2080–2099, RCP 8.5;

[Figure]

Fig. 5 Mean predicted annual water table of 20 GCMs in five wetlands in the Southeastern United States (unit: cm).

Note: Bseline:1980–1999, historical baseline period;  Mid 20-4.5:2040–2059, RCP 4.5; Late 20-4.5:2080–2099, RCP 4.5;  Mid 21-8.5:2040–2059, RCP 8.5, Late 21-8.5:2080–2099, RCP 8.5;

**Point #13**

**COMMENT:** Line 685: Figure 6 – These axes should be flipped for ease of interpretation. Also fix the legend so it doesn't look like it was drawn by hand. Consider changing the x-axis label and putting the site name in the figure panel itself.

RESPONSE: We much appreciate the reviewer's careful review. The Figure 6 was revised with fixing the axes, legend, label, and site names. Also, the figure quality was improved in the revised version in line 760 as follows.

[Figure]

[Figure]

Fig. 6 Exceedance probability of the mean predicted water table in the growing season of 20 GCMs in five wetlands in the Southeastern United States (unit: cm).

Note: Bseline:1980–1999, historical baseline period;

Mid 20-4.5:2040–2059, RCP 4.5; Late 20-4.5:2080–2099, RCP 4.5;

Mid 21-8.5:2040–2059, RCP 8.5, Late 21-8.5:2080–2099, RCP 8.5;

---

## Author Response (AR1)

**GENERAL COMMENTS:** This study uses a linear regression model to estimate future water table changes in five forest sites in Southeastern US. The topic is interesting. However, this study lacks of innovation in climate change impact assessment. Besides the simple regression model, there are critical issues/errors in the methodology (see major comments). Please find my detailed comments below.

**RESPONSE:** This study develops site-specific empirical hydrologic models for five representative forested wetlands with different characteristics by synthesizing long-term observed meteorological and hydrological data and coupling climate changes from 20 CGMs. These wetlands represent typical Cypress Ponds/Swamps, Carolina Bays, Pine Flatwoods, and Wet Pine, and natural Bottomland Hardwoods ecosystems, and cover a range of climatic/topographic gradients and different management conditions located in the SE US. This study provides quantitative information on the different potential magnitudes of wetland hydrological responses to future climate changes for adaptive management of typical forested wetlands in the southern U.S. Based on the Reviewer's suggestions, we have added clarifications in the methodology in the revised manuscript, made relevant corrections and improvements in the title, main body, tables, figures, and also references (details see the marked version).

**MAJOR COMMENTS:**

**Point #1**

**COMMENT:** In fitting the regression model, water table at the current time step is the dependent variable while water table at the previous time step is included as one of the independent variables. This is not reasonable given the potential autocorrelations between current water table and antecedent water table especially at the daily scale. In fact, the unbelievably high R2 value of 0.97 in predicting water table at the LP site could be due to the inclusion of antecedent water table in the statistical model. What's more, how can future water table be predicted by the statistical model which requires inputs of antecedent water table conditions?

**RESPONSE:** We address reviewer's three major questions below.
(1) Reasonability of model structure using the antecedent water table as an explanatory variable,

(i) This study adopted the well-established methodology of dynamic panel model widely used in statistics and econometrics. The dynamic panel modeling includes the first lag dependent variable coupled with the explanatory variables (e.g. P, PET, in this study). The model structure with a given lag effect were successfully used in the previous studies for hydroregime prediction (Greenberg et al., 2015, Webb et al., 2003), urban water demand prediction (Almendarez-Hernández et al., 2016; Arbués et al., 2004; Arbues et al., 2010, Lyman, 1992), and energy-food-water interaction modelling (Liu et al., 2017; Ozturk, 2015). Lyman's (1992) and Ozturk (2015) confirmed the adjustments significance of minimizing heterogeneity in the traditional Ordinary Least Squares assumptions by including the first lagged dependent variable. Webb et al. (2003) improved the sensitivity and explanatory power of the hourly based water-air temperature regression models by incorporating a lagged response of water temperature. In a wetland hydrology and climate change study, Greenberg et al. (2015) successfully forecasting the hydroregimes of multiple wetlands by modeling the water table depth using water level of the prior week and precipitation as predictors of current water table.

(ii) The statistical model structure has physical meaning and can be viewed from the perspective of water balance. A lagged effect of water table was supported by the water balances and the water table dynamics of wetlands. It is due to the fact that wetland groundwater has memories which can be carried beyond the next season as to influence the water balance in the coming years (Miguez-Macho and Fan 2012). Firstly, the water balance of the five selected wetlands can be written as $WT_t - WT_{t-1} = \alpha + \beta(P_t - ET_t) + \varepsilon_t$. It indicates the causality balances between water tables changes (left side terms of the above equation i.e., $WT_t - WT_{t-1}$) and its water

availability changes (right side terms of the above equation). The item of $WT_{t-1}$ depending on the time scale can be considered as the memory effect of water tables. By moving the memory item ($WT_{t-1}$) to the right side, the different statistical coefficient of $WT_{t-1}$ can reflect the different memory characters/effects in the five selected wetlands. Based on these reasons, we believe the model structure that includes a lagged water table in this study can offer more information concerning not only water tables changes due to changes in climate variables, but also the different memory effects of different wetlands in this region.

(iii) All the information is not contained in the antecedent water table conditions of wetlands. The lagged water table only offers the basis for the current condition, however, the forcings (e.g. climate and water availability in this study) alters the water table depth.  In another word, the water table depth would always decrease/increase along with a given initial discharge/recharge condition for the wetland. But, a wetland actually alternatively discharges or recharges for the flatness, thus water tables fluctuated with the forcings. Therefore, both the forcings and the lagged water table would determine the water tables for a wetland. What's more, when using the water table at LP site with forcings from FL-WET site, the statistics show that in spite of the same antecedent condition, the $R^2$ becomes poorer to 0.54 from 0.83.

(iv) From the perspective of the proved wide-sense stationary first-order autoregressive process of water tables in the five selected wetlands, the variance will not change with the autoregressive process introduced into the statistic model. For an autoregressive process given by: $Y_t = \alpha + \beta Y_{t-1} + \varepsilon_t$ , where $\varepsilon_t$ is a white noise process with zero mean and constant variance $\sigma_\varepsilon^2$. The first-order autoregressive process is wide-sense stationary when and only when $|\beta|<1$, because it is the output of a stable filter with a white noise input (Mills, 1990). In the wetlands analyzed in the study, the coefficient of $WT_{t-1}$ are<1 (Table 3), indicating a wide-sense stationary autoregressive process of water table. Thus the variance of the process does not change with simulation over time, and the intertemporal effect ($\beta^n \varepsilon_1$ ) of shocks diminishes toward zero in the limit.

(v) From the perspective of the independence of the explanatory variables, introduce of antecedent water table does not violate the independence requirements among the explanatory variables. Independence between the explanatory variables was satisfied since correlation coefficient between P-PET and $WT_{t-1}$ was very poor (<2.7) for the five selected wetlands in the study.

According to the comment, we added the justification in the revised manuscript (Line 161-166, similarly hereinafter) to make it clear and concise for readers.

(2) For the question of "unbelievably high R2 value of 0.97 at LP site", it was actually the correlation coefficient (R) for model verification, which was corrected in line 250-252. The model was developed with a determination coefficient ($R^2$, proportion of the variance in predicted water table) of 0.83, which was also the highest among the five wetlands. It appears during the verification, the model was able to well capture the variations for the entire verification period for the LP site. The good capability of LP site may be explained by the contribution of the high interception value of the statistical model, which may include the contribution of artificial drainage system of this pine plantation. As mentioned in (1)-iii, all the information is not contained in the antecedent water table conditions of LP site. The $R^2$ becomes poorer to 0.54 from 0.83 when using the forcings from FL-WET site, in spite of the same antecedent condition.

(3) For the question of 'daily scale', this regression model is developed at the 15-day time step, not on a daily scale (Line 174, Line 178-179).

(4) The related new references were added to the revised manuscript as follows:

"Almendarez-Hernández, M., Avilés Polanco, G., Hernández Trejo, V., Ortega-Rubio, A., and Beltrán Morales, L.: Residential Water Demand in a Mexican Biosphere Reserve: Evidence of the Effects of Perceived Price, Water, 8, 428, 2016. (Line 432-436)

Arbués, F., Barberán, R., and Villanúa, I.: Price impact on urban residential water demand: A dynamic panel data approach, Water Resour Res, 40, 2004. (Line 442-443)

Arbues, F., Garcia-Valinas, M. A., and Villanua, I.: Urban Water Demand for Service and Industrial Use: The Case of Zaragoza, Water Resour Manag, 24, 4033-4048, 2010. (Line 444-445)

Liu, G., Yang, Z., Tang, Y., and Ulgiati, S.: Spatial correlation model of economy-energy-pollution interactions: The role of river water as a link between production sites and urban areas, Renewable and Sustainable Energy Reviews, 69, 1018-1028, 2017. (Line 542-544)

Lyman, R. A.: Peak and off-peak residential water demand, Water Resour Res, 28, 2159-2167, 1992. (Line 554)

Mills, Terence C. Time Series Techniques for Economists. Cambridge University Press, 1990. (Line 568)

Ozturk, I.: Sustainability in the food-energy-water nexus: Evidence from BRICS (Brazil, the Russian Federation, India, China, and South Africa) countries, Energy, 93, 999-1010, 2015. (Line 595-596)

Webb, B., Clack, P., and Walling, D.: Water–air temperature relationships in a Devon river system and the role of flow, Hydrological processes, 17, 3069-3084, 2003. " (Line 656-657)

**Point #2**

**COMMENT:** There are major issues related to the short calibration and validation periods for the statistical model. For example, two years of data is used for fitting regression model for the AR site while one year is used for validation. I am wondering whether climatic conditions in the validation year is significant different from the calibration year? Future climate especially for the later periods of 21st century would be quite different from the calibration periods based on which the regression model is constructed. Therefore, the historical relations trained from such a short time period may not hold in the future with significant changes in climate.

**RESPONSE:** We agree that regression models are limited data availability and can contribute to model deficiency (Line 195). However, long-term, high resolution observed wetland water table data for multiple sites in the southeast U.S. are extremely rare. For example, the Alligator River Wildlife Refuge bottom hardwood wetland (AR site), is located in a remote location and water table data are in the only data sets extremely valuable to characterize the local hydrological conditions. Fortunately, the dataset covered both dry and wet years at the selected sites and was ideal for model development and validation purposes. For example, at wetlands FL–UP and FL-WET, the time series including wet and dry years (1993–1994) was used to develop the model, and the remaining data (1992, 1995, and 1996) were used for model validation (Fig. 3). Additionally, the model was then applied to predict water table based on the GCMs dataset in a full time scale (1950-2099) including both the baseline period (1980-1999) and the future period (2040-2059, 2080-2099) (Line 141-151). Thus, during model applied to predict water table based on the GCMs dataset, future and historical climate will share the same bias. The changes from the historical to the future are comparable.

**Point #3**

**COMMENT:** The downscaled GCM climate should be validated for the baseline period in the study sites before it can be used for future predictions.

**RESPONSE:** The climate data that we used in baseline period (1980-1999) and future period (2040-2059, 2080-2099) are both the downscaled GCM climate dataset (Line 141-151). These downscaled climate datasets are found to be as a good match

(90% of Perkins PDF skill score between 0.8-0.95) regionally over the southeastern United States by means observations, and the entire distribution of observations (Keellings, 2016). Besides, both the baseline period and future period would share the same bias. Thus, the hydrologic and climate changes from the baseline period to the end of this century are comparable. According to the comment, we added a few sentences (Line 135-137 and Line 141-151) in the revised manuscript.

The related new reference was added in the revised manuscript as follows:
"Keellings, D.: Evaluation of downscaled CMIP5 model skill in simulating daily
    maximum temperature over the southeastern United States, Int J Climatol, 36,
    4172-4180, 2016." (Line 529-530)

**MINOR COMMENT:**

**Point #1**

**COMMENT:** In Section 80, RCP stands for "Representative Concentration Pathway" rather than "Regional Concentration Pathways".

**RESPONSE:** Corrected in line 77. Thanks.

**Point #2**

**COMMENT:** Hamon's equation is selected for estimating PET. Justifications on this should be added.

**RESPONSE:** The justifications were added in line 167-170 as follows:
"PET is mainly controlled by net radiation, air temperature, wind speed, and air humidity (Hargreaves and Samani, 1982). Due to data availability, this study used the air temperature-based Hamon equation to calculate PET (Hamon, 1963). The Hamon's PET method has been widely used worldwide to estimate potential forest water use (Sun et al., 2002)."

The new reference was added to the revised manuscript as follows:
"Hargreaves, G. H. and Samani, Z. A.: Estimating potential evapotranspiration,
    Journal of the Irrigation and Drainage Division, 108, 225-230, 1982." (Line 515-
    516)

**Point #3**

**COMMENT:** In section 130, the estimated PET is adjusted to match "realistic" PET values for forests. What are the realistic PET values for forests?

**RESPONSE:** The 'realistic PET' was a typo; it should be the actual ET for the forests in this region. The sentence was re-written as "A correction coefficient (Sun et al., 2002) was used to adjust PET calculated by Hamon's equation to better represent the forest PET for the study region. The correction coefficients for North Carolina ranged from 1.0 to 1.2 (Federer and Lash, 1978b), and was 1.3 for the Florida site (Sun et al., 1998). To be consistent and reduce uncertainty of PET estimates, 1.2 was used for all five wetlands in this study." in line 125-128.

**Point #4**

**COMMENT:** The climate for the baseline period is based on observations or GCM simulations?

**RESPONSE:** We appreciate the reviewer's careful review. The climate for the baseline period 1980–1999 (historical run) was based on of the downscaled GCM datasets (1950-2099) (Line 141- 151).

**Point #5**

**COMMENT:** A table with a brief description of the GCMs should be added.

**RESPONSE:** A new table (Supplementary Table S1) with summary of the GCMs was added in the revised version.

**Table S1 Summary of the 20 CMIP5 GCMs used in this study from the downscaled MACA dataset.**

| No. | Model Name | Country | Model Institution | Atmosphere Resolution (Lon x Lat) |
|-----|------------|---------|-------------------|------------------------------------|
| 1 | bcc-csm1-1 | China | Beijing Climate Center, China Meteorological Administration | 2.8 deg x 2.8 deg |
| 2 | bcc-csm1-1-m | China | Beijing Climate Center, China Meteorological Administration | 1.12 deg x 1.12 deg |
| 3 | BNU-ESM | China | College of Global Change and Earth System Science, Beijing Normal University, China | 2.8 deg x 2.8 deg |
| 4 | CanESM2 | Canada | Canadian Centre for Climate Modeling and Analysis | 2.8 deg x 2.8 deg |
| 5 | CCSM4 | USA | National Center of Atmospheric Research, USA | 1.25 deg x 0.94 deg |
| 6 | CNRM-CM5 | France | National Centre of Meteorological Research, France | 1.4 deg x 1.4 deg |
| 7 | CSIRO-Mk3-6-0 | Australia | Commonwealth Scientific and Industrial Research Organization/Queensland Climate Change Centre of Excellence, Australia | 1.8 deg x 1.8 deg |
| 8 | GFDL-ESM2M | USA | NOAA Geophysical Fluid Dynamics Laboratory, USA | 2.5 deg x 2.0 deg |
| 9 | GFDL-ESM2G | USA | NOAA Geophysical Fluid Dynamics Laboratory, USA | 2.5 deg x 2.0 deg |
| 10 | HadGEM2-ES | United Kingdom | Met Office Hadley Center, UK | 1.88 deg x 1.25 deg |
| 11 | HadGEM2-CC | United Kingdom | Met Office Hadley Center, UK | 1.88 deg x 1.25 deg |
| 12 | inmcm4 | Russia | Institute for Numerical Mathematics, Russia | 2.0 deg x 1.5 deg |
| 13 | IPSL-CM5A-LR | France | Institut Pierre Simon Laplace, France | 3.75 deg x 1.8 deg |
| 14 | IPSL-CM5A-MR | France | Institut Pierre Simon Laplace, France | 2.5 deg x 1.25 deg |
| 15 | IPSL-CM5B-LR | France | Institut Pierre Simon Laplace, France | 2.75 deg x 1.8 deg |
| 16 | MIROC5 | Japan | Atmosphere and Ocean Research Institute (The University of Tokyo), National Institute for Environmental Studies,and Japan Agency for Marine-Earth Science and Technology | 1.4 deg x 1.4 deg |
| 17 | MIROC-ESM | Japan | Japan Agency for Marine-Earth Science and Technology, Atmosphere and Ocean Research Institute (The University of Tokyo), and National Institute for Environmental Studies | 2.8 deg x 2.8 deg |
| 18 | MIROC-ESM-CHEM | Japan | Japan Agency for Marine-Earth Science and Technology, Atmosphere and Ocean Research Institute (The University of Tokyo), and National Institute for Environmental Studies | 2.8 deg x 2.8 deg |
| 19 | MRI-CGCM3 | Japan | Meteorological Research Institute, Japan | 1.1 deg x 1.1 deg |
| 20 | NorESM1-M | Norway | Norwegian Climate Center, Norway | 2.5 deg x 1.9 deg |

**Point #6**
COMMENT: This study focus on the projection of water table depth at five forest sites. The title should mention "water table depth" rather than "hydrology" which is a broad concept.

RESPONSE: The word "hydrology" was changed to "water table level" in the title.

**(2) Comments from anonymous referee #2 and author's response:**

**RESPONSES TO REVIEWERS' COMMENTS**

We are grateful to the reviewers and associate editor for their detailed and insightful comments. In light of the suggestions, we have made efforts to significantly revise the manuscript. The suggestions and comments have substantially contributed towards improving the paper. More details are as follows:
**GENERAL COMMENTS:** The discussion paper uses an empirical approach to determine hydrological effects on climate change for 5 wetland types in the southeastern U.S. The paper generally has scientific significance in that it tries to address uncertainties associated with climate change, and the overall structure of the paper is clear and concise. However, the paper lacks rigorous evaluation of the model and results. The general model structure appears flawed (see comments below), and model results are seemingly taken at face value. For example, the authors do not address a major source of uncertainty associated with climate change: water use efficiency (WUE). Climate change is associated with increases in CO2, not just temperature, and increased CO2 is known to increase WUE, which would have major implications for the results presented here. There was also little consideration given to changes in water availability for vegetation, which drives actual ET. Additionally, the graphics and tables were lacking in quality and readability, and should be revised. There were many typographical and grammatical errors. Overall, this paper needs major revisions.

**RESPONSE:** We are very thankful for the reviewer's detailed reviews about the uncertainty associated with our modeling results related to other factors of future climate change, e.g. the increased WUE because of the increased $CO_2$, and the changes in water availability for vegetation. This work focuses on the wetland groundwater variability due

to climate drivers' change such as precipitation and air temperature. We did not specifically consider the effects of increased $CO_2$ on vegetation growth and productivity which may further affect wetland hydrology in the study due to the following reasons. First, some of the GCM models used here already contain a dynamics vegetation model (e.g., Yu et al., 2016); thus, vegetation responses to increased $CO_2$ have been considered in these GCMs. Second, previous results suggest that precipitation and surface air temperature are the two first order climate variables to drive wetland groundwater (Liljedahl et al., 2011). Third, quantitative analysis about the wetland evaporation/evapotranspiration comparison found that transpiration from canopy provided few contributions to evapotranspiration (Li et al., 2009). We thank the reviewer to get our attention and will study the aspect in our future work although this is out of the scope of work of the present study. According to the comment, we added recommendations for future work in line 386-387 (in the revised manuscript, similarly hereinafter). The related references were as follows:

Li, Y. J., Zhou, L., Xu, Z. Z., and Zhou, G. S.: Comparison of water vapour, heat and energy exchanges over agricultural and wetland ecosystems, Hydrological Processes, 23, 2069-2080, 2009.

Liljedahl, A. K., Hinzman, L. D., Harazono, Y., Zona, D., Tweedie, C. E., Hollister, R. D., Engstrom, R., and Oechel, W. C.: Nonlinear controls on evapotranspiration in arctic coastal wetlands, Biogeosciences, 8, 3375-3389, 2011.

Yu, M., G. Wang, and H. Chen, Quantifying the impacts of land surface schemes and dynamic vegetation on the model dependency of projected changes in surface energy and water budgets, J. Adv. Model. Earth Syst., 8, 370–386, 2016.

We have also improved the methodology, results, and discussion re-plotted the graphs using high resolution and re-done the tables to improve the quality and readability as suggested (details see the marked version).

**SCIENTIFIC COMMENTS:**

For the reviewer's convenience during re-review, we numbered his/her comments and included our corresponding responses, below.

**Point #1**

**COMMENT:** Line 78-79: Why is Greenberg et al. (2015) referenced here without discussing how it "satisfactorily" used an empirical model? All you say is that they used one.

**RESPONSE:** We much appreciate the reviewer's careful review. We further discussed the references in line 70-73 as follows:
"Greenberg et al. (2015) developed an empirical model and demonstrated its utility for climate-change planning by forecasting the weekly hydrologic regimes from 2012 to 2060 and examining the indirect impacts of climate change on biological diversity."

**Point #2**

**COMMENT**: Line 158-159: It is not clear what the rationale is for using lagged water table as an independent variable? It seems clear that the most recent water table value will be highly correlated to the water table now. Is this just using autocorrelation as a covariate? Consider revision.

**RESPONSE:** The lagged water table was actually considered as a covariate. The rationale to use the lagged water table was to account for its contribution to the current water table, in addition to the role of current climate and water availability, based on the considerations as follows:

(i) This study adopted the well-established methodology of dynamic panel model widely used in statistics and econometrics. The dynamic panel modeling includes the first lag dependent variable coupled with the explanatory variables (e.g. P, PET, in this study). The model structure with a given lag effect were successfully used in the previous studies for hydroregime prediction (Greenberg et al., 2015, Webb et al., 2003), urban water demand prediction (Almendarez-Hernández et al., 2016; Arbués et al., 2004; Arbues et al., 2010, Lyman, 1992), and energy-food-water interaction modelling (Liu et al., 2017; Ozturk, 2015). Lyman's (1992) and Ozturk (2015) confirmed the adjustments significance of minimizing heterogeneity in the traditional Ordinary Least Squares assumptions by including the first lagged dependent variable. Webb et al. (2003) improved the sensitivity and explanatory power of the hourly based water-air temperature regression models by incorporating a lagged response of water temperature. In a wetland hydrology and climate change study, Greenberg et al. (2015) successfully forecasting the

hydroregimes of multiple wetlands by modeling the water table depth using water level of the prior week and precipitation as predictors of current water table.

(ii) The statistical model structure has physical meaning and can be viewed from the perspective of water balance. A lagged effect of water table was supported by the water balances and the water table dynamics of wetlands. It is due to the fact that wetland groundwater has memories which can be carried beyond the next season as to influence the water balance in the coming years (Miguez-Macho and Fan 2012). Firstly, the water balance of the five selected wetlands can be written as $WT_t - WT_{t-1} = \alpha + \beta(P_t - ET_t) + \varepsilon_t$. It indicates the causality balances between water tables changes (left side terms of the above equation i.e., $WT_t - WT_{t-1}$) and its water availability changes (right side terms of the above equation). The item of $WT_{t-1}$ depending on the time scale can be considered as the memory effect of water tables. By moving the memory item ($WT_{t-1}$) to the right side, the different statistical coefficient of $WT_{t-1}$ can reflect the different memory characters/effects in the five selected wetlands. Based on these reasons, we believe the model structure that includes a lagged water table in this study can offer more information concerning not only water tables changes due to changes in climate variables, but also the different memory effects of different wetlands in this region.

(iii) All the information is not contained in the antecedent water table conditions of wetlands. The lagged water table only offers the basis for the current condition, however, the forcings (e.g. climate and water availability in this study) alters the water table depth. In another word, the water table depth would always decrease/increase along with a given initial discharge/recharge condition for the wetland. But, a wetland actually alternatively discharges or recharges for the flatness, thus water tables fluctuated with the forcings. Therefore, both the forcings and the lagged water table would determine the water tables for a wetland. What's more, when using the water table at LP site with forcings from FL-WET site, the statistics show that in spite of the same antecedent condition, the $R^2$ becomes poorer to 0.54 from 0.83.

(ⅳ) From the perspective of the proved wide-sense stationary first-order autoregressive process of water tables in the five selected wetlands, the variance will not change with the autoregressive process introduced into the statistic model. For an autoregressive process given by: $Y_t = \alpha + \beta Y_{t-1} + \varepsilon_t$, where $\varepsilon_t$ is a white noise process with zero mean and constant variance $\sigma_\varepsilon^2$. The first-order autoregressive process is widesense stationary when and only when $|\beta|<1$, because it is the output of a stable filter with a white noise input (Mills, 1990). In the wetlands analyzed in the study, the coefficient of $WT_{t-1}$ are<1 (Table 3), indicating a wide-sense stationary autoregressive process of water table. Thus the variance of the process does not change with simulation over time, and the intertemporal effect ($\beta^n \varepsilon_1$) of shocks diminishes toward zero in the limit.

(ⅴ)From the perspective of the independence of the explanatory variables, introduce of antecedent water table does not violate the independence requirements among the explanatory variables. Independence between the explanatory variables was satisfied since correlation coefficient between P-PET and $WT_{t-1}$ was very poor (<2.7) for the five selected wetlands in the study.

According to the comment, the justification was further concisely added in line 161-166 in the revised manuscript to make it clearer for readers. Also, the related new references were added to the revised manuscript as follows:

"Almendarez-Hernández, M., Avilés Polanco, G., Hernández Trejo, V., Ortega-Rubio, A., and Beltrán Morales, L.: Residential Water Demand in a Mexican Biosphere Reserve: Evidence of the Effects of Perceived Price, Water, 8, 428, 2016. (Line 432-436)

Arbués, F., Barberán, R., and Villanúa, I.: Price impact on urban residential water demand: A dynamic panel data approach, Water Resour Res, 40, 2004. (Line 442-443)

Arbues, F., Garcia-Valinas, M. A., and Villanua, I.: Urban Water Demand for Service and Industrial Use: The Case of Zaragoza, Water Resour Manag, 24, 4033-4048, 2010. (Line 444-445)

Liu, G., Yang, Z., Tang, Y., and Ulgiati, S.: Spatial correlation model of economy-energy-pollution interactions: The role of river water as a link between production sites and urban areas, Renewable and Sustainable Energy Reviews, 69, 1018-1028, 2017. (Line 542-544)

Lyman, R. A.: Peak and off-peak residential water demand, Water Resour Res, 28, 2159-2167, 1992. (Line 554)

Mills, Terence C. Time Series Techniques for Economists. Cambridge University Press, 1990. (Line 568)

Ozturk, I.: Sustainability in the food-energy-water nexus: Evidence from BRICS (Brazil, the Russian Federation, India, China, and South Africa) countries, Energy, 93, 999-1010, 2015. (Line 595-596)

Webb, B., Clack, P., and Walling, D.: Water–air temperature relationships in a Devon river system and the role of flow, Hydrological processes, 17, 3069-3084, 2003. " (Line 656-657)

**Point #3**

COMMENT: Line 160: It seems like an autocorrelation covariance structure should be used given the time-series nature of the data.

RESPONSE: We view that "an autocorrelation covariance structure" here as "covariance structure of autocorrelation". For an autoregressive process given by: $Y_t = \alpha + \beta Y_{t-1} + \varepsilon_t$, where $\varepsilon_t$ is a white noise process with zero mean and constant variance $\sigma_\varepsilon^2$. The autocovariance is $B_n = E(X_{t+n}X_t) - \mu^2 = \frac{\sigma_\varepsilon^2}{1-\beta^2}\beta^{|n|}$, where $\mu$ is the model mean, and n is the time step (Mills, 1990). In this study, autoregressive process is wide-sense stationary ($|\beta|<1$), thus $\frac{\sigma_\varepsilon^2}{1-\beta^2}\beta^{|n|}$ diminishes toward zero in the limit in the general model form (Line 180). Besides, the autocorrelation nature for the given data time-series was first tested to select the final variables. The autocorrelation of water tables with lag time of 0 day, 15 days, 30 days, 45 days, and 60 days were tested, as well as their correlations with the other explanatory variables. Consequently, in the final model (line 217), only the 15-day (one time step) lagged water table was chose as an explanatory variable for the best statistical results. Thus only a constant variance $\sigma_\varepsilon^2$ exists in the final selected wide-sense stationary first order autoregressive process.

**Point #4**

COMMENT: Line 176: Water loss is also controlled by net groundwater flow, but more importantly by vegetation access to water and vegetation water use efficiency (WUE), which are not accounted for in the model. And because we know that WUE is strongly influenced by CO2 concentrations, this appears to be a major deficiency in the model.

RESPONSE: Please see our replies to the General Comment (page 1-2). We agree with reviewer on the hydrological processes identified. However, given the empirical nature of our model, these factors are not explicitly considered. We have also added some discussion regarding the potential uncertainty originating from discounting the effects of increase in $CO_2$ on WUE in the revised version (see lines 386-387):
"For example, an increase in atmospheric CO2 concentration is likely to increase plant water use efficiency and thus the ET and water balance of wetlands (Brummer et al., 2012)"

The related new reference was added to the revised manuscript as follows:

"Brummer, C., Black, T. A., Jassal, R. S., Grant, N. J., Spittlehouse, D. L., Chen, B., Nesic, Z., Amiro, B. D., Arain, M. A., Barr, A. G., Bourque, C. P. A., Coursolle, C., Dunn, A. L., Flanagan, L. B., Humphreys, E. R., Lafleur, P. M., Margolis, H. A., McCaughey, J. H., and Wofsy, S. C.: How climate and vegetation type influence evapotranspiration and water use efficiency in Canadian forest, peatland and grassland ecosystems, Agricultural and Forest Meteorology, 153, 14-30, 2012." (Line 457-461)

**Point #5**

COMMENT: Line 186-195: Did you also test for assumptions of normality of residuals and homoscedasticity of residuals? If you did not take into account autocorrelation of covariance it is likely that these assumptions may be violated.

RESPONSE: The normality and the homoscedasticity for both the five sites were tested before the models were applied to the prediction. The residual plots of the five specific models showed that errors are homoscedastic, and both the residuals and the normal probability plot showed the normality of the residuals in the five selected wetlands. With introducing the autoregressive variable, the Durbin's h also indicated the autocorrelation disturbance process. We also added the clarification to the revised version in line 191, and line 220-221:

"Also, the autocorrelation disturbance process was tested by Durbin's *h* statistic (*Bhargava et al.*, 1982)." (Line 191)

"Durbin's h statistic showed that all five wetland regressions support the autocorrelation disturbance process." The related figures of AR site were shown as an example as follows: (Line 220-221)

[Figure]

[Figure]

[Figure]

Figure 1 Residuals and normal probability plots of AR site, a) P-PET residual plot, b) WTt-1 residual plot, c) normal probability plot. The residuals and standard residuals of the observed Y (water table) have a mean of zero ($6 \times 10^{-16}$, $1 \times 10^{-16}$, respectively).

**Point #6**

**COMMENT**: Lines 196-199: What did you find with Durbin's h? Did it support autocorrelation or not?

**RESPONSE:** The Durbin's h tests for the models showed that all the five wetlands regression models support the autocorrelation, by comparing the lower and upper critical value from the Durbin-Watson Table. The results of DW test (2.59 for AR, 2.66 for LP, 1.32 for FL-UP, 2.63 for FL-WET, and 1.49 for SC) showed that all five wetland regressions support the autocorrelation disturbance process. We also added the statement to the revised manuscript in line 220-221.

**Point #7**

**COMMENT**: Line 234-236: Are these estimates for changes to PET based purely on temperature changes? This seems important to note.

**RESPONSE:** Yes, PET is estimated based on air temperature only in addition to day length following Sun et al., (2002). The associated note was added in line 167-170 as follows:
"PET is mainly controlled by net radiation, air temperature, wind speed, and air humidity (Hargreaves and Samani, 1982). Due to data availability, this study used the air temperature-based Hamon equation to calculate PET (Hamon, 1963). The Hamon's PET method has been widely used worldwide to estimate potential forest water use (Sun et al., 2002)."

The new reference was added to the revised manuscript as follows:

"Hargreaves, G. H. and Samani, Z. A.: Estimating potential evapotranspiration, Journal of
the Irrigation and Drainage Division, 108, 225-230, 1982." (Line 515-516)

**Point #8**

**COMMENT**: Line 270-271: This sentence is the opposite of what is suggested by the
figure and is confusing to interpret.

**RESPONSE:** The sentence was revised in line 278-282 as follows;
"Additionally, all the predicted 15-day water table levels were negative (i.e., water table
< 0 cm) at LP, FL–UP, and SC, meaning there would be no surface water ponding in the
RCP 8.5 scenario, as well as in the baseline scenario (Table 5, Fig. 6). In contrast, the
wetlands AR and FL–WET show a lower probability (i.e., 40 % for FL–WET, 49 % for
AR) with no surface water ponding in the baseline, but a significantly increasing
probability of 62 % and 93%, respectively, in the RCP 8.5 scenario."

**Point #9**

**COMMENT**: Lines 272-279: This section is very difficult to understand, especially
trying to reconcile with figures. Suggest re-writing.

**RESPONSE:** The section has been re-written (Line 283-290) as follows:
"Despite the fact that LP, FL–UP and SC were all predicted to have no surface water
(water table < 0 cm) over the study period, the soil saturation status (water table depth
still within 30 cm) varied by location (Table 5). Site LP and FL–UP would completely
dry up by 2099 based on the RCP 8.5 scenario. Wetland SC was saturated 100 % of the
time during the baseline period, but the saturation period would decrease to 57 % by
2099. The wetland FL-WET would be the most sensitive of the five sites. In FL-WET, the
probability would increase most in losing surface water ponding (increasing from 40% to
93 % from the baseline period to 2099) and decrease most in saturated soil (decreasing
from 100% to 63 %). Notably, the wetland AR would be the only wetland that would

remain 100 % saturated under all future scenarios including RCP 8.5 scenario (Table 5, Fig. 6)."

**Point #10**

**COMMENT**: Lines 283-285: Where are the R2 values coming from? Are these ratios of R2 to other sites? Clarification needed.

**RESPONSE:** The $R^2$ values are the coefficient of determination of the regressions model results. The clarification was added in the revised version in line 293-296 as follows:

"The lower R2 values (0.69) of the model in FL-UP site than the FL-WET site (0.78) might be caused by other impacts beyond the model considerations, e.g. the hydrologic interaction between the uplands and the wetlands in the Florida site. Also, the temporal scale of 15 days may better capture the hydrological changes in FL-UP rather than FL-WET due to a faster drainage system in the FL-UP site."

**Point #11**

**COMMENT**: Line 288: Did you statistically test that the model coefficients were similar? They do not seem too similar to me…

**RESPONSE:** We much appreciate the reviewer's careful review. The sentence was deleted in the revised version.

**Point #12**

**COMMENT**: Where is the discussion of how the model did not perform well? The model appears to be much flashier and tends to overpredict relative to observed data? RMSE or some other metric would be useful as a comparison.

**RESPONSE:** We have addressed reviewer's concerns by adding RMSE values. The discussion was added in line 231-237 and line 195 as follows:

"The average water table was over-predicted by 1.4 cm for LP (-106.25 cm for observation, -104.85 cm for prediction, with root mean square error (RMSE) of 4.92 cm, similarly hereinafter), 0.95 cm for FL-WET (19.02 cm, 19.97 cm, with RMSE of 9.23 cm), and 1.3 cm for SC (-19.1 cm, -17.8 cm, with RMSE of 5.16 cm). Also, it was under-predicted by 2.11 cm for FL-UP (-48.97 cm, -51.08 cm, with RMSE of 5.9 cm), and 0.38 cm for AR (-4.19 cm, -4.57 cm, with RMSE of 3.71 cm). The models captured the changing water table level even during an extremely dry year (e.g. 2007-2008 at LP). For the FL-WET, the water table levels were over-predicted in the normal period while the observations and the predictions matched better during the dry year in 1993. Overall, …" (line 231-237).

"Limited data availability can contribute to model deficiency." (Line 195).

**TECHNICAL CORRECTIONS:**

**Point #1**

**COMMENT**: Line 56: "… and more powerful hurricanes landfall." Word choice here is awkward.

**RESPONSE:** In line 49, the phrase was revised to "more intense Atlantic hurricanes".

**Point #2**

**COMMENT**: Line 58: "process-based study" should be "process-based studies".

**RESPONSE:** The phrase was revised to "process-based studies" in the updated version.

**Point #3**

**COMMENT**: Line 70: add "and" before "…their potential uses…"

**RESPONSE:** The "and" was accordingly added.

**Point #4**

**COMMENT**: Line 73-75: This sentence needs revision for clarity and grammar.

**RESPONSE:** The sentence was revised in line 65-67 as follows:
"Conversely, in spite of the weakness of assumption of static relationships between climate and hydrological response patterns in the future, statistical models have advantages of both high efficiency and acceptable performance when applied over multiple sites."

**Point #5**

**COMMENT**: Line 75: "Performance such type of models…" a word is missing.

**RESPONSE:** The sentence was revised as "Especially, performance of such empirical models…" in line 67-68.

**Point #6**

**COMMENT**: Line 84: change "increased" to "subsequent increases".

**RESPONSE:** The word was changed accordingly in line 68.

**Point #7**

**COMMENT**: Line 88: There is an extra "s" after the parentheses.

**RESPONSE:** The extra "s" was deleted in the revised version.

**Point #8**

**COMMENT**: Line 289: change "higher" to "lower".

**RESPONSE:** The word was accordingly changed to "lower".

**Point #9**

**COMMENT**: Line 387: Missing a word in "Climate change from single has been used…" and "wetalnd" is misspelled.

**RESPONSE:** The sentence was revised as "Climate data from single GCMs (Greenberg et al., 2015; Wang et al., 2015) have been used in wetland hydrological response, …" in line 404-405.

**Point #10**

**COMMENT**: Line 625: Table 1 should have consistent formatting for each of the data in columns for ease of comparison. Consider a more generic description of soils instead of series names.

**RESPONSE:** The Table 1 was reformatted and revised in line 670, and a more generic description of soils instead of series names were used for the sites. The climate data from different time series including the most of the observation years is to better reflect different climate background during model development.

**Point #11**

**COMMENT**: Line 670: Figure 3(d) what is meant by the orange dots?

**RESPONSE:** Figure 3 (line 705, as follows) was updated in the revised version with mistake fixed and the site names put in the figure panel itself. We also improved the quality of Figure 1 (line 690, as follows) and Figure 2 (line 700, as follows).

[Figure]

Fig. 3 Scatter plots of the observed and predicted mean water table in five wetlands in the Southeastern United States (unit: cm), Dashed lines are 1:1 line.

[Figure]

Fig. 1 Study area, where the star symbol marks the study site location. Wetland AR: wetland of Alligator River National Wildlife Refuge in North Carolina; wetland LP: wetland of loblolly pine plantation in North Carolina; wetland SC: wetland in South Carolina; wetlands in Florida: wetland FL–UP (upland in Florida) and FL–WET.

[Figure]

Fig. 2 Comparison of observed and simulated 15-day water table in five wetlands in the Southeastern United States, WT is water table.

**Point #12**

**COMMENT**: Line 680 and 685: Figures 4 and 5 are begging to have significance letters attributed to each boxplot.

**RESPONSE:** We much appreciate the reviewer's careful review. The Figures 4 and 5 were revised in line 710 (as follows) and line 720 with significance letters, site names in the panel itself, and the quality was improved as well.

[Figure]

**Fig. 4 Total annual precipitation minus potential evapotranspiration of 20 GCMs in five wetlands in the southeastern United States (unit: mm).**

Note: Baseline: 1980–1999, historical run of GCMs;

Mid 20-4.5: 2040–2059, RCP 4.5; Late 20-4.5:2080–2099, RCP 4.5;

Mid 21-8.5: 2040–2059, RCP 8.5, Late 21-8.5:2080–2099, RCP 8.5;

[Figure]

**Fig. 5 Mean predicted annual water table of 20 GCMs in five wetlands in the southeastern United States (unit: cm).**

Note: Baseline: 1980–1999, historical run of GCMs;

Mid 20-4.5: 2040–2059, RCP 4.5; Late 20-4.5:2080–2099, RCP 4.5;

Mid 21-8.5: 2040–2059, RCP 8.5, Late 21-8.5:2080–2099, RCP 8.5;

**Point #13**

**COMMENT:** Line 685: Figure 6 – These axes should be flipped for ease of interpretation. Also fix the legend so it doesn't look like it was drawn by hand. Consider changing the x-axis label and putting the site name in the figure panel itself.

RESPONSE: We much appreciate the reviewer's careful review. The Figure 6 was revised with fixing the axes, legend, label, and site names. Also, the figure quality was improved in the revised version in line 730 as follows.

[revised manuscript text omitted]

Various hydrological models, ranging from regression models to complex distributed models, have been used to study hydrological response to climate change. For example, the physically based distributed model MIKE SHE has been applied to forested wetlands in the SE US (Dai et al., 2010; Lu et al., 2009; House et al., 2016). The hydrological regime of wetland forests on the coastal plains of South Carolina was found to be highly sensitive to annual precipitation and temperature changes (Dai et al., 2010). The water table of pine flatwoods in Florida was predicted to be 20-40 cm lower than that of a baseline scenario when precipitation decreased by 10 % or temperature increased by 2 °C (Lu et al., 2009).

Integrated studies on the impacts of climate change on multiple wetlands in the SE US are limited. Physically based hydrological models provide a refined understanding of hydrologic processes (Yu et al., 2015; Chen et al., 2015) and quantification of hydrologic states and fluxes (Qu and Duffy, 2007; Shen and Phanikumar, 2010). However, these models are generally data (Bhatt et al., 2014) and computation intensive (Vivoni et al., 2011), and their potential uses are often undercut by equifinality of parameters (*Beven*, 1993; *Kumar et al.*, 2013; *Pokhrel et al.*, 2008). Implementing distributed hydrologic models across multiple wetlands that cover a range of climatic, topographic, and management conditions is challenging due to the computational expense, lack of fine scale input data, and difficulty in application for multiple sites (*Grayson et al.*, 1992). Conversely, in spite of the weakness of assumption of static relationships between climate and hydrological response patterns in the future, statistical models have advantages of both high efficiency and acceptable performance when applied over multiple sites. The performance of empirical models in climate change studies appears to be powerful when incorporating downscaled Global Climate Model (GCM) outputs (*Sachindra et al.*, 2013; *Li et al.*, 2016). For example, *Li et al.* (2016) used log-linear models for 21 rainfall stations and severn hydrometric stations to predict hydrological drought. Greenberg et al. (2015) developed an empirical model and demonstrated its utility for climate-change planning by forecasting

Field Code Changed …
Field Code Changed …
Field Code Changed …
Formatted …
Field Code Changed …
Field Code Changed …
Field Code Changed …
Field Code Changed …
Field Code Changed …
Field Code Changed …

[revised manuscript text omitted]

**(4) Author's changes in supplementary materials**

**Supplementary Materials**

Table list:

Table S1 Summary of the 20 CMIP5 GCMs used in this study from the downscaled MACA dataset.

Table S2 Annual changes of climate variables of 20 GCMs for WT, P-PET, P, PET, and AT in wetland AR.

Table S3 Annual changes of climate variables of 20 GCMs for WT, P-PET, P, PET, and AT in wetland LP.

Table S4 Annual changes of climate variables of 20 GCMs for WT, P-PET, P, PET, and AT in wetland SC.

Table S5 Annual  changes of climate variables of 20 GCMs for WT, P-PET, P, PET, and AT in FL–UP.

Table S6 Annual changes averages of climate variables of 20 GCMs for WT, P-PET, P, PET, and AT in FL–WET.

Figure list:

Fig. S1 Mean annual air temperature of 20 GCMs (unit: Deg C).

Fig. S2 Mean annual PET of 20 GCMs (unit: mm).

Fig. S3 Total annual precipitation of 20 GCMs (unit: mm).

**Tables and Figures**

Table S1 Summary of the 20 CMIP5 GCMs used in this study from the downscaled MACA dataset.

| No. | Model Name | Country | Model Institution | Atmosphere Resolution (Lon x Lat) |
|-----|-----------|---------|-------------------|-----------------------------------|
| 1 | bcc-csm1-1 | China | Beijing Climate Center, China Meteorological Administration | 2.8 deg x 2.8 deg |
| 2 | bcc-csm1-1-m | China | Beijing Climate Center, China Meteorological Administration | 1.12 deg x 1.12 deg |
| 3 | BNU-ESM | China | College of Global Change and Earth System Science, Beijing Normal University, China | 2.8 deg x 2.8 deg |
| 4 | CanESM2 | Canada | Canadian Centre for Climate Modeling and Analysis | 2.8 deg x 2.8 deg |
| 5 | CCSM4 | USA | National Center of Atmospheric Research, USA | 1.25 deg x 0.94 deg |
| 6 | CNRM-CM5 | France | National Centre of Meteorological Research, France | 1.4 deg x 1.4 deg |
| 7 | CSIRO-Mk3-6-0 | Australia | Commonwealth Scientific and Industrial Research Organization/Queensland Climate Change Centre of Excellence, Australia | 1.8 deg x 1.8 deg |
| 8 | GFDL-ESM2M | USA | NOAA Geophysical Fluid Dynamics Laboratory, USA | 2.5 deg x 2.0 deg |
| 9 | GFDL-ESM2G | USA | NOAA Geophysical Fluid Dynamics Laboratory, USA | 2.5 deg x 2.0 deg |
| 10 | HadGEM2-ES | United Kingdom | Met Office Hadley Center, UK | 1.88 deg x 1.25 deg |
| 11 | HadGEM2-CC | United Kingdom | Met Office Hadley Center, UK | 1.88 deg x 1.25 deg |
| 12 | inmcm4 | Russia | Institute for Numerical Mathematics, Russia | 2.0 deg x 1.5 deg |
| 13 | IPSL-CM5A-LR | France | Institut Pierre Simon Laplace, France | 3.75 deg x 1.8 deg |
| 14 | IPSL-CM5A-MR | France | Institut Pierre Simon Laplace, France | 2.5 deg x 1.25 deg |
| 15 | IPSL-CM5B-LR | France | Institut Pierre Simon Laplace, France | 2.75 deg x 1.8 deg |
| 16 | MIROC5 | Japan | Atmosphere and Ocean Research Institute (The University of Tokyo), National Institute for Environmental Studies,and Japan Agency for Marine-Earth Science and Technology | 1.4 deg x 1.4 deg |
| 17 | MIROC-ESM | Japan | Japan Agency for Marine-Earth Science and Technology, Atmosphere and Ocean Research Institute (The University of Tokyo), and National Institute for Environmental Studies | 2.8 deg x 2.8 deg |
| 18 | MIROC-ESM-CHEM | Japan | Japan Agency for Marine-Earth Science and Technology, Atmosphere and Ocean Research Institute (The University of Tokyo), and National Institute for Environmental Studies | 2.8 deg x 2.8 deg |
| 19 | MRI-CGCM3 | Japan | Meteorological Research Institute, Japan | 1.1 deg x 1.1 deg |
| 20 | NorESM1-M | Norway | Norwegian Climate Center, Norway | 2.5 deg x 1.9 deg |

**Table S2** Annual changes of climate variables of 20 GCMs for WT, P-PET, P, PET, and AT in site AR.

| Scenario | WT mean (cm/365d) | WT min (cm/365d) | P-PET (mm/365d) | P (mm/365d) | PET (mm/365d) | AT (Deg C/365d) |
|---|---|---|---|---|---|---|
| B | 0 | -1 | 290 | 1266 | 977 | 16.5 |
| F1 | -1 | -3 | 233 | 1295 | 1062 | 18.1 |
| F2 | -1 | -3 | 215 | 1322 | 1107 | 18.9 |
| F3 | -2 | -3 | 199 | 1298 | 1100 | 18.8 |
| F4 | -4 | -5 | 106 | 1303 | 1198 | 20.4 |
| ΔF1B | -1 | -2 | -57 | 29 | 85 | 1.6 |
| ΔF2B | -1 | -2 | -75 | 56 | 130 | 2.4 |
| ΔF3B | -2 | -2 | -91 | 32 | 123 | 2.3 |
| ΔF4B | -4 | -4 | -184 | 37 | 221 | 3.9 |

Note: B:1980-1999, historical baseline; F1:2040-2059, RCP 4.5, future scenario 1; F2:2080-2099, RCP 4.5, future scenario 2; F3:2040-2059, RCP 8.5, future scenario 3; F4:2080-2099, RCP 8.5, future scenario 4; Values of ΔFnB (n=1, 2, 3, 4) indicate the values of scenario Fn minus values of baseline scenario.

**Table S2** Annual changes of climate variables of 20 GCMs for WT, P-PET, P, PET, and AT in site LP.

| Scenario | WT mean (cm/365d) | WT min (cm/365d) | P-PET (mm/365d) | P (mm/365d) | PET (mm/365d) | AT (Deg C/365d) |
|---|---|---|---|---|---|---|
| B | -100 | -110 | 313 | 1275 | 963 | 16.1 |
| F1 | -106 | -116 | 263 | 1318 | 1055 | 17.9 |
| F2 | -107 | -118 | 241 | 1343 | 1103 | 18.7 |
| F3 | -108 | -118 | 231 | 1325 | 1093 | 18.5 |
| F4 | -119 | -127 | 138 | 1338 | 120 | 20.4 |
| ΔF1B | -6 | -6 | -50 | 43 | 92 | 1.8 |
| ΔF2B | -7 | -8 | -72 | 68 | 140 | 2.6 |
| ΔF3B | -8 | -8 | -82 | 50 | 130 | 2.4 |
| ΔF4B | -19 | -17 | -175 | 63 | 238 | 4.3 |

Note: B:1980-1999, historical baseline; F1:2040-2059, RCP 4.5, future scenario 1; F2:2080-2099, RCP 4.5, future scenario 2; F3:2040-2059, RCP 8.5, future scenario 3; F4:2080-2099, RCP 8.5, future scenario 4; Values of ΔFnB (n=1, 2, 3, 4) indicate the values of scenario Fn minus values of baseline scenario.

**Table S3 Annual changes of climate variables of 20 GCMs for WT, P-PET, P, PET, and AT in site SC.**

| Scenario | WT mean (cm/365d) | WT min (cm/365d) | P-PET (mm/365d) | P (mm/365d) | PET (mm/365d) | AT (Deg C/365d) |
|---|---|---|---|---|---|---|
| B | -16 | -19 | 142 | 1192 | 1050 | 18.1 |
| F1 | -18 | -22 | 67 | 1217 | 1150 | 19.8 |
| F2 | -18 | -21 | 60 | 1262 | 1202 | 20.6 |
| F3 | -19 | -22 | 49 | 1241 | 1192 | 20.5 |
| F4 | -23 | -25 | -65 | 1252 | 1316 | 22.4 |
| B | -16 | -19 | 142 | 1192 | 1050 | 18.1 |
| ΔF2B | -2 | -2 | -82 | 70 | 152 | 2.5 |
| ΔF3B | -3 | -3 | -93 | 49 | 142 | 2.5 |
| ΔF4B | -7 | -6 | -207 | 60 | 266 | 4.3 |

Note: B:1980-1999, historical baseline; F1:2040-2059, RCP 4.5, future scenario 1; F2:2080-2099, RCP 4.5, future scenario 2; F3:2040-2059, RCP 8.5, future scenario 3; F4:2080-2099, RCP 8.5, future scenario 4; Values of ΔFnB (n=1, 2, 3, 4) indicate the values of scenario Fn minus values of baseline scenario.

**Table S4 Annual changes of climate variables of 20 GCMs for WT, P-PET, P, PET, and AT in FL–UP.**

| Scenario | WT mean (cm/365d) | WT min (cm/365d) | P-PET (mm/365d) | P (mm/365d) | PET (mm/365d) | AT (Deg C/365d) |
|---|---|---|---|---|---|---|
| B | -73 | -81 | 165 | 1318 | 1153 | 20.6 |
| F1 | -78 | -88 | 83 | 1333 | 1250 | 22.2 |
| F2 | -78 | -83 | 74 | 1376 | 1302 | 22.9 |
| F3 | -80 | -88 | 46 | 1338 | 1292 | 22.8 |
| F4 | -90 | -99 | -124 | 1297 | 1420 | 24.6 |
| ΔF1B | -5 | -7 | -82 | 15 | 97 | 1.6 |
| ΔF2B | -5 | -2 | -91 | 58 | 149 | 2.3 |
| ΔF3B | -7 | -7 | -119 | 20 | 139 | 2.2 |
| ΔF4B | -17 | -18 | -289 | -21 | 267 | 4.0 |

Note: B:1980-1999, historical baseline; F1:2040-2059, RCP 4.5, future scenario 1; F2:2080-2099, RCP 4.5, future scenario 2; F3:2040-2059, RCP 8.5, future scenario 3; F4:2080-2099, RCP 8.5, future scenario 4;

Values of ΔFnB (n=1, 2, 3, 4) indicate the values of scenario Fn minus values of baseline scenario.

**Table S5 Annual changes in averages of future climate variables of 20 GCMs for WT, P-PET, P, PET, and AT in FL–WET.**

| Scenario | WT mean (cm/365d) | WT min (cm/365d) | P-PET (mm/365d) | P (mm/365d) | PET (mm/365d) | AT (Deg C/365d) |
|---|---|---|---|---|---|---|
| B | 2 | -6 | 165 | 1318 | 1153 | 20.6 |
| F1 | -4 | -14 | 83 | 1333 | 1250 | 22.2 |
| F2 | -5 | -10 | 74 | 1376 | 1302 | 22.9 |
| F3 | -7 | -17 | 46 | 1338 | 1292 | 22.8 |
| F4 | -20 | -30 | -124 | 1297 | 1420 | 24.6 |
| ΔF1B | -6 | -8 | -82 | 15 | 97 | 1.6 |
| ΔF2B | -7 | -4 | -91 | 58 | 149 | 2.3 |
| ΔF3B | -9 | -11 | -119 | 20 | 139 | 2.2 |
| ΔF4B | -22 | -24 | -289 | -21 | 267 | 4.0 |

Note: B:1980-1999, historical baseline; F1:2040-2059, RCP 4.5, future scenario 1; F2:2080-2099, RCP 4.5, future scenario 2; F3:2040-2059, RCP 8.5, future scenario 3; F4:2080-2099, RCP 8.5, future scenario 4;

Values of ΔFnB (n=1, 2, 3, 4) indicate the values of scenario Fn minus values of baseline scenario.

[Figure]

[Figure]

**Fig. S1 Mean annual air temperature of 20 GCMs (unit: Deg C)**

¶

F1:2040-2059, RCP 4.5, future scenario 1; F2:2080-2099, RCP 4.5, future scenario 2; ¶
F3:2040-2059, RCP 8.5, future scenario 3; F4:2080-2099, RCP 8.5, future scenario 4;¶
¶
------------------------------Page Break------------------------------
¶
¶                                                                    …

[Figure]

Fig. S2 Mean annual PET of 20 GCMs (unit: mm).

F1:2040-2059, RCP 4.5, future scenario 1; F2:2080-2099, RCP 4.5, future scenario 2; ¶
F3:2040-2059, RCP 8.5, future scenario 3; F4:2080-2099, RCP 8.5, future scenario 4;¶
¶
—————————Page Break—————————
¶
¶

[Figure]

**Fig. S3 Total annual precipitation of 20 GCMs (unit: mm)**

F1:2040-2059, RCP 4.5, future scenario 1; F2:2080-2099, RCP 4.5, future scenario 2; ¶
F3:2040-2059, RCP 8.5, future scenario 3; F4:2080-2099, RCP 8.5, future scenario 4;¶

---

## Author Response (AR2)

**REPLIES TO the 2ⁿᵈ REVIEWERS ON THE REVISED MANUSCRIPT "Modeling the Potential Impacts of Climate Change on the Hydrology of Selected Forested Wetlands in the Southeastern United States" by Zhu et al.**

We thank the reviewer #2 for his/her insightful comments and we have implemented the suggestions as described in detail below.

**Reply to Review #2:**

**GENERAL:** *Authors have shown great efforts in revising the manuscript; however, there are still some minor problems.*

**TECHNICAL CORRECTIONS:**
*1. For the "Supplementary Table S1", the name of the last category (e.g., Atmosphere Resolution (lat x lon)) can be changed into "Atmosphere Grid in Degrees (lat x lon)"; thus, the subsequent column can be written as, for example, 2.8 x 2.8 only.*

**RESPONSE:** We have changed the name of the last category and the subsequent column of the "Supplementary Table S1" as suggested.

*2. Please clarify/specify the phrase "high efficiency and acceptable performance" of the statistical models; for example, in which aspect do they have high efficiency and acceptable performance?*

**RESPONSE:** We have rewritten the phrase to make it clearer as following "high efficiency in computation and acceptable performance in modelling when applied over multiple sites" in the revised manuscript (line 69, in the marked version).

*3. "single GCMs" or "single GCM"? "GCM" or "GCMs"? please keep them consistent throughout the manuscript.*

**RESPONSE:** Both "single GCM" and "GCMs" are used to refer a global climate model, and multi models (plural). We have corrected the singular (such as "single GCM" in line 71, 145, 153, 175, 497, and 505) and plural forms (e.g. line 502, and 404) in the revised manuscript to keep them consistent.

*4. Plots for Figure 3 are not the same size; also, the positions of site name and R are not aligned, and please unify the format of whether a space is needed before/after the sign "=" for the R value.*

**RESPONSE:** We replotted Figure 3 to make sure that all plots are the same size, the position of site name and R are aligned with same formats.

*5. Same problem for Figure 2 and Figure 4, where the site names need to be kept in one common place. If it is not possible to keep the names in the upper-left corner within the plot, authors can consider putting the names outside the plots on the left side, for example. OR just name them with a,b,c,d, etc and specify them within the figure captions.*

**RESPONSE:** We replotted Figures 2 and 4 in the main text, Figures S1, S2, and S3 in the Supplementary Material according to the comment.

*6. It seems that the map in figure 1 is distorted. It is better to keep the correct size of the map.*

**RESPONSE:** We replotted Figure 1 as suggested.

*7. For Figure 6, the site name has already been mentioned on each of the graphs, so there is no need to re-write the site name on the y-axis. Moreover, the site names are not in alignment with each other, please adjust.*

**RESPONSE:** Figure 6 was accordingly revised.

*8. Please change "Representative Concentration Pathway (RCP) 4.5 and RCP 8.5 scenarios" to "Representative Concentration Pathways (RCPs) 4.5 and 8.5 scenarios". Please correct them in the rest of your manuscript accordingly.*

**RESPONSE:** The phrase was accordingly revised in line 26, as well as in the rest manuscript, e.g. line87, 142, 177-180, 275-360, 416.

*9. In your title, you changed "hydrology" to "water table content".*

**RESPONSE:** According to the comment, we changed the title from "Modeling the Potential Impacts of Climate Change on the Hydrology of Selected Forested Wetlands in the Southeastern United States" to "Modeling the Potential Impacts of Climate Change on

the Water Table Level of Selected Forested Wetlands in the Southeastern United States", since the **water table level** was the actual hydrological indicator used in this study.

*10. In line 116, please change "severn" to "seven".*

**RESPONSE:** We corrected the typo. Thanks.

*11.Since you changed "hydrology" to "water table content" in your title to specify your research, please justify in your manuscript why you still use the word "wetland hydrology" since it is a broad concept.*

**RESPONSE:** Changed in the manuscript.

*12. Line 250: please change "... include the precipitation ..." to " ... include precipitation…".*

**RESPONSE:** Changed

*13: Line 252: instead of putting the link as in-text citation, please give a name of the author/institution of this website. Links can be included in the references.*

**RESPONSE:** According to the comment, we have added the name of the institution, "The United States Naval Observatory (USNO)" (line 127), and the website "http://aa.usno.navy.mil/data/docs/Dur_OneYear.php" (line 755) in the revised manuscript.

*14. Lines 267: please change "mean daily" to "daily mean", and "the 20 GCMs" to "20 GCMs".*

**RESPONSE:** Changed.

*15: Line 274: Please rewrite the sentence "we analyzed the historical and future climate conditions key to wetland hydrology, including the daily maximum temperature near the surface (2 m), daily minimum temperature near the surface (2 m),…". Authors can consider changing the "daily maximum temperature near the surface (2m)" to "daily maximum near surface temperature" since near surface temperature refers to "temperature at 2 meter".*

**RESPONSE:** We have changed the sentence from "...including the daily maximum temperature near the surface (2m), daily minimum temperature near the surface (2 m)…" to "…including the daily maximum near surface (2 m) temperature, daily minimum near surface (2 m) temperature…" (Line 149-150) according to the comment. Thank you.

*16. In the manuscript text, authors have mentioned that 1980-1999 refers to the end of 20th century, 2040-2059 as mid-21st century, and 2080-2099 for the end of 21st century; while for Figure 4, the captions were "mid 20-4.5: 2040-2059, RCP 4.5" and "late 20-4.5: 2080-2099, RCP 4.5". Please justify.*

**RESPONSE:** According to the comment, we have corrected the caption of Figure 4 as following, "Baseline is 1980–1999, historical run of GCMs; mid21 is 2040–2059, under RCPs 4.5 and 8.5 scenarios; late21 is 2080–2099, under RCPs 4.5 and 8.5 scenarios." We also corrected the captions of and replotted Figures 5 and 6 in the main text and Figures S1, S2, and S3 in the Supplementary Material.

*17: Line 551: please change "Future predicted PET" to "The predicted PET".*

**RESPONSE:** Changed (line 300).

*18. Line 562: "RCP4.5" and "RCP8.5", please keep them consistent in the manuscript since in other places authors have written as "RCP 4.5" and "RCP 8.5" with a space in-between.*

**RESPONSE:** According to the comment, we now use "RCP 4.5" and "RCP 8.5" with a space in-between throughout the entire revised paper.

*19: Line 562: please change "Fig. 5" to "Figure 5"; also keep them consistent throughout the manuscript for other figures.*

**RESPONSE:** The words were accordingly changed and kept consistent throughout the manuscript for all figures.

*20: Line 628: please clarify the sentence "Although the model structure proved the same in all five wetlands, and had good performance overall, a closer comparison shows differing influences on the wetland hydrology". What do you mean by "proved the same"?*

**RESPONSE:** We have rewritten the sentence as "Although the statistical model follows a similar structure, i.e., including the same two explanatory variables in all five wetlands, and is proved to have good simulation performance overall, a closer comparison of the modelled water table levels among the five wetlands shows different climate influences" (line 405-406).

*21. Please change "in RCP 8.5 scenario" to "under RCP 8.5 scenario", the same applied to RCP 4.5, and please keep them consistent throughout the manuscript.*

**RESPONSE:** The phrases were changed accordingly and kept consistent throughout the manuscript, e.g. line 275, 277, 301-306, 850, 860, 895.

*22. Please rewrite the sentence "This may not only be due to the PET increase, which was similar to that of the other three sites (AR, LP, and SC), but also because the precipitation decreased in the wetland FL, while it increased at the other sites.". It seems a little bit wordy and not clear/concise enough.*

**RESPONSE:** We have rewritten the sentences as "For the other three sites (AR, LP, and SC), this may be due to large increase in PET. Moreover, the precipitation decreases in the wetland FL, while it increases at other sites." (line 381-383).

*23. Please be careful when using the word "the". There are some places where "the" can be eliminated. Proofread by a native speaker is recommended.*

**RESPONSE:** We much appreciate the reviewer's careful review. After the proofread, some places where "the" was eliminated, e.g. line 126, 141, 382, 410, 411, 427.

*24. Line 723: please justify why your "models were able to accurately predict different water table dynamics…"? Please also noted that "the" can be eliminated for "the different water table dynamics".*

**RESPONSE:** We much appreciate the reviewer's comment and added the justification in the revised manuscript: "However, according to the model performance and results analysis in this study (Table 2, Figures 2 and 3), our models were proved to be able to well predict different water table dynamics under a range of climatic and management conditions across the SE US region." in line 425-428.

*25. Line 912, please change "the model developed by this study" to "the model developed in this study".*

**RESPONSE:** The word was accordingly changed to "the models developed in this study" in Line 482.

**26.** *Line 973, please change "under future climate change" to "under future climate change scenarios".*

**RESPONSE:** The word was accordingly changed to "under future climate change scenarios" in line 512.

*27. The conclusion section is a little bit short. Please strengthen this part by including what the authors have done and what are the significant results, the associated contributions, as well as the future work if possible.*

**RESPONSE:** According to the comment, we have rewritten the conclusion section by including our research work and significant results (line 509-532) as following:

[revised manuscript text omitted]

Note: Baseline:

Mid 20-4.5:

Mid 21-8.5: 2040–2059, RCP

895

**Supplementary Materials**

Table list:

Table S1 Summary of the 20 CMIP5 GCMs used in this study from the downscaled MACA dataset.

Table S2 Annual changes of climate variables of 20 GCMs for WT, P-PET, P, PET, and AT in wetland AR.

Table S3 Annual changes of climate variables of 20 GCMs for WT, P-PET, P, PET, and AT in wetland LP.

Table S4 Annual changes of climate variables of 20 GCMs for WT, P-PET, P, PET, and AT in wetland SC.

Table S5 Annual changes of climate variables of 20 GCMs for WT, P-PET, P, PET, and AT in FL–UP.

Table S6 Annual changes averages of climate variables of 20 GCMs for WT, P-PET, P, PET, and AT in FL–WET.

Figure list:

Fig. S1 Mean annual air temperature of 20 GCMs (unit: Deg C).

Fig. S2 Mean annual PET of 20 GCMs (unit: mm).

Fig. S3 Total annual precipitation of 20 GCMs (unit: mm).

**Tables and Figures**

**Table S1 Summary of the 20 CMIP5 GCMs used in this study from the downscaled MACA dataset.**

| No. | Model Name | Country | Model Institution | Atmosphere Grid in Degrees (lat x lon) |
|---|---|---|---|---|
| 1 | bcc-csm1-1 | China | Beijing Climate Center, China Meteorological Administration | 2.8 x 2.8 |
| 2 | bcc-csm1-1-m | China | Beijing Climate Center, China Meteorological Administration | 1.12 x 1.12 |
| 3 | BNU-ESM | China | College of Global Change and Earth System Science, Beijing Normal University, China | 2.8 x 2.8 |
| 4 | CanESM2 | Canada | Canadian Centre for Climate Modeling and Analysis | 2.8 x 2.8 |
| 5 | CCSM4 | USA | National Center of Atmospheric Research, USA | 1.25 x 0.94 |
| 6 | CNRM-CM5 | France | National Centre of Meteorological Research, France | 1.4 x 1.4 |
| 7 | CSIRO-Mk3-6-0 | Australia | Commonwealth Scientific and Industrial Research Organization/Queensland Climate Change Centre of Excellence, Australia | 1.8 x 1.8 |
| 8 | GFDL-ESM2M | USA | NOAA Geophysical Fluid Dynamics Laboratory, USA | 2.5 x 2.0 |
| 9 | GFDL-ESM2G | USA | NOAA Geophysical Fluid Dynamics Laboratory, USA | 2.5 x 2.0 |
| 10 | HadGEM2-ES | United Kingdom | Met Office Hadley Center, UK | 1.88 x 1.25 |
| 11 | HadGEM2-CC | United Kingdom | Met Office Hadley Center, UK | 1.88 x 1.25 |
| 12 | inmcm4 | Russia | Institute for Numerical Mathematics, Russia | 2.0 x 1.5 |

**Formatted Table**

| 13 | IPSL-CM5A-LR | France | Institut Pierre Simon Laplace, France | 3.75 x 1.8 |
| 14 | IPSL-CM5A-MR | France | Institut Pierre Simon Laplace, France | 2.5 x 1.25 |
| 15 | IPSL-CM5B-LR | France | Institut Pierre Simon Laplace, France | 2.75 x 1.8 |
| 16 | MIROC5 | Japan | Atmosphere and Ocean Research Institute (The University of Tokyo), National Institute for Environmental Studies,and Japan Agency for Marine-Earth Science and Technology | 1.4 x 1.4 |
| 17 | MIROC-ESM | Japan | Japan Agency for Marine-Earth Science and Technology, Atmosphere and Ocean Research Institute (The University of Tokyo), and National Institute for Environmental Studies | 2.8 x 2.8 |
| 18 | MIROC-ESM-CHEM | Japan | Japan Agency for Marine-Earth Science and Technology, Atmosphere and Ocean Research Institute (The University of Tokyo), and National Institute for Environmental Studies | 2.8 x 2.8 |
| 19 | MRI-CGCM3 | Japan | Meteorological Research Institute, Japan | 1.1 x 1.1 |
| 20 | NorESM1-M | Norway | Norwegian Climate Center, Norway | 2.5 x 1.9 |

**Table S2 Annual changes of climate variables of 20 GCMs for WT, P-PET, P, PET, and AT in site AR.**

| Scenario | WT mean (cm/365d) | WT min (cm/365d) | P-PET (mm/365d) | P (mm/365d) | PET (mm/365d) | AT (Deg C/365d) | |
|---|---|---|---|---|---|---|---|
| B | 0 | -1 | 290 | 1266 | 977 | 16.5 | ← Formatted Table |
| F1 | -1 | -3 | 233 | 1295 | 1062 | 18.1 | |
| F2 | -1 | -3 | 215 | 1322 | 1107 | 18.9 | |
| F3 | -2 | -3 | 199 | 1298 | 1100 | 18.8 | |
| F4 | -4 | -5 | 106 | 1303 | 1198 | 20.4 | |
| ΔF1B | -1 | -2 | -57 | 29 | 85 | 1.6 | |
| ΔF2B | -1 | -2 | -75 | 56 | 130 | 2.4 | |
| ΔF3B | -2 | -2 | -91 | 32 | 123 | 2.3 | |
| ΔF4B | -4 | -4 | -184 | 37 | 221 | 3.9 | |

Note: B:1980-1999, historical baseline; F1:2040-2059, RCP 4.5, future scenario 1; F2:2080-2099, RCP 4.5, future scenario 2; F3:2040-2059, RCP 8.5, future scenario 3; F4:2080-2099, RCP 8.5, future scenario 4; Values of ΔFnB (n=1, 2, 3, 4) indicate the values of scenario Fn minus values of baseline scenario.

**Table S2 Annual changes of climate variables of 20 GCMs for WT, P-PET, P, PET, and AT in site LP.**

| Scenario | WT mean (cm/365d) | WT min (cm/365d) | P-PET (mm/365d) | P (mm/365d) | PET (mm/365d) | AT (Deg C/365d) | |
|---|---|---|---|---|---|---|---|
| B | -100 | -110 | 313 | 1275 | 963 | 16.1 | ← Formatted Table |
| F1 | -106 | -116 | 263 | 1318 | 1055 | 17.9 | |
| F2 | -107 | -118 | 241 | 1343 | 1103 | 18.7 | |
| F3 | -108 | -118 | 231 | 1325 | 1093 | 18.5 | |
| F4 | -119 | -127 | 138 | 1338 | 120 | 20.4 | |
| ΔF1B | -6 | -6 | -50 | 43 | 92 | 1.8 | |
| ΔF2B | -7 | -8 | -72 | 68 | 140 | 2.6 | |
| ΔF3B | -8 | -8 | -82 | 50 | 130 | 2.4 | |
| ΔF4B | -19 | -17 | -175 | 63 | 238 | 4.3 | |

Note: B:1980-1999, historical baseline; F1:2040-2059, RCP 4.5, future scenario 1; F2:2080-2099, RCP 4.5, future scenario 2; F3:2040-2059, RCP 8.5, future scenario 3; F4:2080-2099, RCP 8.5, future scenario 4; Values of ΔFnB (n=1, 2, 3, 4) indicate the values of scenario Fn minus values of baseline scenario.

**Table S3 Annual changes of climate variables of 20 GCMs for WT, P-PET, P, PET, and AT in site SC.**

| Scenario | WT mean (cm/365d) | WT min (cm/365d) | P-PET (mm/365d) | P (mm/365d) | PET (mm/365d) | AT (Deg C/365d) |
|---|---|---|---|---|---|---|
| B | -16 | -19 | 142 | 1192 | 1050 | 18.1 |
| F1 | -18 | -22 | 67 | 1217 | 1150 | 19.8 |
| F2 | -18 | -21 | 60 | 1262 | 1202 | 20.6 |
| F3 | -19 | -22 | 49 | 1241 | 1192 | 20.5 |
| F4 | -23 | -25 | -65 | 1252 | 1316 | 22.4 |
| B | -16 | -19 | 142 | 1192 | 1050 | 18.1 |
| ΔF2B | -2 | -2 | -82 | 70 | 152 | 2.5 |
| ΔF3B | -3 | -3 | -93 | 49 | 142 | 2.5 |
| ΔF4B | -7 | -6 | -207 | 60 | 266 | 4.3 |

Note: B:1980-1999, historical baseline; F1:2040-2059, RCP 4.5, future scenario 1; F2:2080-2099, RCP 4.5, future scenario 2; F3:2040-2059, RCP 8.5, future scenario 3; F4:2080-2099, RCP 8.5, future scenario 4; Values of ΔFnB (n=1, 2, 3, 4) indicate the values of scenario Fn minus values of baseline scenario.

**Table S4 Annual changes of climate variables of 20 GCMs for WT, P-PET, P, PET, and AT in FL–UP.**

| Scenario | WT mean (cm/365d) | WT min (cm/365d) | P-PET (mm/365d) | P (mm/365d) | PET (mm/365d) | AT (Deg C/365d) |
|---|---|---|---|---|---|---|
| B | -73 | -81 | 165 | 1318 | 1153 | 20.6 |
| F1 | -78 | -88 | 83 | 1333 | 1250 | 22.2 |
| F2 | -78 | -83 | 74 | 1376 | 1302 | 22.9 |
| F3 | -80 | -88 | 46 | 1338 | 1292 | 22.8 |
| F4 | -90 | -99 | -124 | 1297 | 1420 | 24.6 |
| ΔF1B | -5 | -7 | -82 | 15 | 97 | 1.6 |
| ΔF2B | -5 | -2 | -91 | 58 | 149 | 2.3 |
| ΔF3B | -7 | -7 | -119 | 20 | 139 | 2.2 |
| ΔF4B | -17 | -18 | -289 | -21 | 267 | 4.0 |

Note: B:1980-1999, historical baseline; F1:2040-2059, RCP 4.5, future scenario 1; F2:2080-2099, RCP 4.5, future scenario 2; F3:2040-2059, RCP 8.5, future scenario 3; F4:2080-2099, RCP 8.5, future scenario 4;

Values of ΔFnB (n=1, 2, 3, 4) indicate the values of scenario Fn minus values of baseline scenario.

**Table S5 Annual changes in averages of future climate variables of 20 GCMs for WT, P-PET, P, PET, and AT in FL–WET.**

| Scenario | WT mean (cm/365d) | WT min (cm/365d) | P-PET (mm/365d) | P (mm/365d) | PET (mm/365d) | AT (Deg C/365d) |
|---|---|---|---|---|---|---|
| B | 2 | -6 | 165 | 1318 | 1153 | 20.6 |
| F1 | -4 | -14 | 83 | 1333 | 1250 | 22.2 |
| F2 | -5 | -10 | 74 | 1376 | 1302 | 22.9 |
| F3 | -7 | -17 | 46 | 1338 | 1292 | 22.8 |
| F4 | -20 | -30 | -124 | 1297 | 1420 | 24.6 |
| ΔF1B | -6 | -8 | -82 | 15 | 97 | 1.6 |
| ΔF2B | -7 | -4 | -91 | 58 | 149 | 2.3 |
| ΔF3B | -9 | -11 | -119 | 20 | 139 | 2.2 |
| ΔF4B | -22 | -24 | -289 | -21 | 267 | 4.0 |

Note: B:1980-1999, historical baseline; F1:2040-2059, RCP 4.5, future scenario 1; F2:2080-2099, RCP 4.5, future scenario 2; F3:2040-2059, RCP 8.5, future scenario 3; F4:2080-2099, RCP 8.5, future scenario 4;

Values of ΔFnB (n=1, 2, 3, 4) indicate the values of scenario Fn minus values of baseline scenario.

Formatted Table

Formatted Table

[Figure]

[Figure]

**Fig. S1 Mean annual air temperature of 20 GCMs (unit: Deg C), where a) is site AR (Alligator River National Wildlife Refuge in North Carolina), b) is site LP (loblolly pine plantation in North Carolina), c) is site FL-UP (upland in Florida) and site FL-WET (wetland in Florida), and d) is site SC (wetland in South Carolina). Baseline is 1980–1999, historical run of GCMs; mid21 is 2040–2059, under RCPs 4.5 and 8.5 scenarios; late21 is 2080–2099, under RCPs 4.5 and 8.5 scenarios.**

[Figure]

**Fig. S2 Mean annual PET of 20 GCMs (unit: mm), where a) is site AR (Alligator River National Wildlife Refuge in North Carolina), b) is site LP (loblolly pine plantation in North Carolina), c) is site FL-UP (upland in Florida) and site FL-WET (wetland in Florida), and d) is site SC (wetland in South Carolina). Baseline is 1980–1999, historical run of GCMs; mid21 is 2040–2059, under RCPs 4.5 and 8.5 scenarios; late21 is 2080–2099, under RCPs 4.5 and 8.5 scenarios.**

[Figure]

**Fig. S3 Total annual precipitation of 20 GCMs (unit: mm), where a) is site AR (Alligator River National Wildlife Refuge in North Carolina), b) is site LP (loblolly pine plantation in North Carolina), c) is site FL-UP (upland in Florida) and site FL-WET (wetland in Florida), and d) is site SC (wetland in South Carolina). Baseline is 1980–1999, historical run of GCMs; mid21 is 2040–2059, under RCPs 4.5 and 8.5 scenarios; late21 is 2080–2099, under RCPs 4.5 and 8.5 scenarios.**